# Learning mutational signatures and their multidimensional genomic properties with TensorSignatures

Harald Vöhringer[1], Arne Van Hoeck [2], Edwin Cuppen[2,3] & Moritz Gerstung [1,4✉]

We present TensorSignatures, an algorithm to learn mutational signatures jointly across different variant categories and their genomic localisation and properties. The analysis of 2778 primary and 3824 metastatic cancer genomes of the PCAWG consortium and the HMF cohort shows that all signatures operate dynamically in response to genomic states. The analysis pins differential spectra of UV mutagenesis found in active and inactive chromatin to global genome nucleotide excision repair. TensorSignatures accurately characterises transcription-associated mutagenesis in 7 different cancer types. The algorithm also extracts distinct signatures of replication- and double strand break repair-driven mutagenesis by APOBEC3A and 3B with differential numbers and length of mutation clusters. Finally, TensorSignatures reproduces a signature of somatic hypermutation generating highly clustered variants at transcription start sites of active genes in lymphoid leukaemia, distinct from a general and less clustered signature of Polη-driven translesion synthesis found in a broad range of cancer types. In summary, TensorSignatures elucidates complex mutational footprints by characterising their underlying processes with respect to a multitude of genomic variables.

[1] European Molecular Biology Laboratory, European Bioinformatics Institute (EMBL-EBI), Hinxton, UK. [2] Center for Molecular Medicine and Oncode Institute, University Medical Center Utrecht, Universiteitsweg 100, Utrecht, The Netherlands. [3] Hartwig Medical Foundation, Amsterdam, The Netherlands. [4] European Molecular Biology Laboratory, Genome Biology Unit, Heidelberg, Germany. ✉email: moritz.gerstung@ebi.ac.uk

Cancer arises through the accumulation of mutations caused by multiple processes that leave behind distinct patterns of mutations on the DNA. A number of studies have analysed cancer genomes to extract such mutational signatures using computational pattern recognition algorithms such as non-negative matrix factorisation (NMF) over catalogues of single nucleotide variants (SNVs) and other mutation types[1–8]. So far, mutational signature analysis has provided more than 50 different single base substitution patterns, indicative of a range of endogenous mutational processes, as well as genetically acquired hypermutation and exogenous mutagen exposures[9].

Mutational signature analysis via computational pattern recognition draws its strength from detecting recurrent patterns of mutations across catalogues of cancer genomes. As many mutational processes also generate characteristic multi nucleotide variants (MNVs)[10,11], insertion and deletions (indels)[12–14], and structural variants (SVs)[6,15–17] it appears valuable to jointly deconvolve broader mutational catalogues to further understand the multifaceted nature of mutagenesis.

Moreover, it has also been reported that mutagenesis depends on a range of additional genomic properties, such as the transcriptional orientation and the direction of replication[18–20], and sometimes manifests as local hypermutation (kataegis)[1]. Additionally, epigenetic and local genomic properties can also influence mutation rates and spectra[21–23]. In fact, these phenomena may help to more precisely characterise the underlying mutational processes, but the large number of possible combinations makes the resulting multidimensional data structure unamenable to conventional matrix factorisation methods.

We present TensorSignatures, a multidimensional tensor factorisation framework incorporating the aforementioned features for a more comprehensive extraction of mutational signatures. We apply TensorSignatures to 2778 whole genomes from the Pan Cancer Analysis of Whole Genomes (PCAWG) consortium[24], and validate our findings in an additional 3824 metastatic cancer whole genomes from the Hartwig Medical Foundation (HMF)[25]. The resulting tensor signatures add considerable detail to known mutational signatures in terms of their genomic determinants and broader mutational context. Strikingly, some signatures are being further subdivided based on genomic properties, illustrating the differential manifestation of the same mutational process in different parts of the genome. This includes UV-mutagenesis and tobacco associated mutations, manifesting at differential rates in active and quiescent chromatin, and enables the algorithm to detect the prevalence of transcription-associated mutagenesis[18,26] in more cancers than currently appreciated. The incorporation of additional variant types enables TensorSignatures to delineate two APOBEC signatures manifesting as replication associated mutations, or highly clustered SV-associated base substitutions indicative of APOBEC3A and 3B[1,18,19,27–30]. Finally, TensorSignatures confirms localised somatic hypermutation at transcription start sites in lymphoid neoplasms[7], with a distinct spectrum from a mostly unclustered, genome-wide signature of translesion synthesis found in a range of other cancer types[28]. Taken together, TensorSignatures sheds light on the manifold influences that underlie mutagenesis and helps to pinpoint mutagenic influences by jointly learning mutation patterns and their genomic determinants. TensorSignatures is implemented using the powerful TensorFlow[31] backend. The software is available as a python package on the PyPI repository and results presented in this study can also be explored on a webserver - see section Code availability at the end of the article.

## Results

**Multidimensional genomic features of mutagenesis.** It is common practise in mutational signature analysis to classify single base substitutions by expressing the mutated base pair in terms of its pyrimidine equivalent (C > A, C > G, C > T, T > A, T > C and T > G) plus the flanking 5′ and 3′ bases. We additionally categorised other mutation types into 91 MNV classes, 62 indel classes, and used the classification of SVs provided by the PCAWG Structural Variants Working Group[17]. In addition to the immediate base context, a number of genomics features have been described to influence mutation rates. Here, we use 5 different genomic annotations—transcription and replication strand orientation, nucleosomal occupancy, epigenetic states as well as clustered hypermutation—and generate 96-dimensional base substitution spectra for each possible combination of these genomic states separately and for each sample. Partitioning variants creates a seven-dimensional count tensor (a multidimensional array), owing to the multitude of possible combinations of different genomic features (Fig. 1a).

Both transcription and replication introduce a strand specificity, which provides a distinction of pyrimidine and purine base context, which is considered to be indistinguishable in the

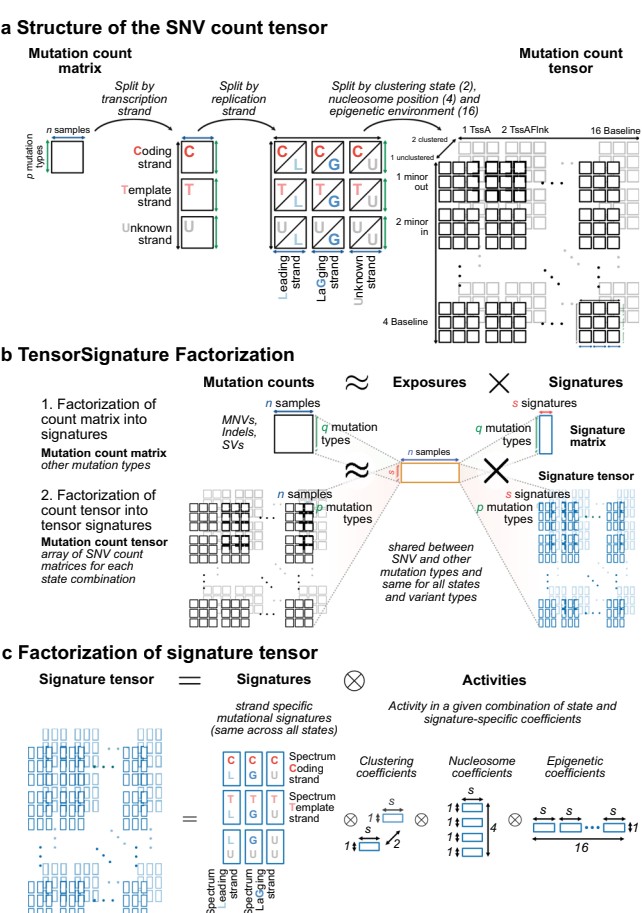

**Fig. 1 A multidimensional tensor factorisation framework to extract mutational signatures. a** Splitting variants by transcriptional and replicational strand, and genomic states creates an array of count matrices, a multidimensional tensor, in which each matrix harbours the mutation counts for each possible combination of genomic states. **b** TensorSignatures factorises a mutation count tensor (SNVs) into an exposure matrix and signature tensor. Simultaneously, other mutation types (MNVs, indels, SVs), represented as a conventional count matrix are factorised using the same exposure matrix. **c** The signature tensor has itself a lower dimensional structure, defined by the product of strand-specific signatures, and coefficients reflecting the activity of the mutational process in a given genomic state combination.

conventional 96-dimensional representation of single base substitution spectra. In transcribed genes, mutation rates may differ between template and coding strand, because RNA polymerase II recruits transcription-coupled nucleotide excision repair (TC-NER) upon lesion recognition on transcribed DNA only. Thus, TC-NER usually leads to lower mutation rates on the template strand, but also the opposite effect—transcription associated mutagenesis (TAM)—occurs[18,26]. Similar effects are observed between leading and lagging strand replication[18,20], possibly because the leading strand is continuously synthesised by DNA polymerase $\epsilon$, while lagging strand DNA synthesis is conducted by DNA polymerase $\delta$, and is discontinuous due to formation of Okazaki fragments. Since not all mutations can be oriented either due to absent or bidirectional transcription, or because of unknown preferred replication direction far from a replication origin, this creates a total of $3 \times 3 =$ (template, coding, unknown) $\times$ (leading, lagging, unknown) combinations of orientation states in the count tensor (Fig. 1a).

Numerous studies found a strong influence of chromatin features on regional mutation rates. Strikingly, these effects range from the 10 bp periodicity on nucleosomes[23] to the scale of kilo- to megabases caused by the epigenetic state of the genome[21]. To understand how mutational processes manifest on histone-bound DNA, we computed the number of variants on minor groove DNA facing away from and towards histone proteins, and linker DNA between two consecutive nucleosomes. Additionally, we utilised ChromHMM annotations from 127 cell-lines[32] to annotate genomic regions with consensus epigenetic states, which we used to assign SNVs to epigenetic contexts. Together this adds two dimensions of size 4 and 16 to the count tensor (Fig. 1a).

Finally, there are mutational processes capable of introducing large numbers of clustered mutations within confined genomic regions. This phenomenon is termed kataegis[1] and is thought to be caused by multiple mutational processes[28]. To detect such mutations, we developed a hidden markov model (HMM) to assign the states clustered and unclustered to each mutation based on the inter-mutation distance between consecutive mutations. Separating clustered from unclustered mutations adds the final dimension in the mutation count tensor, which has a total of 6 dimensions with $2 \times 576 = 1152$ combinations of states (Fig. 1a).

**TensorSignatures learns signatures based on mutation spectra and genomic properties.** At its core, mutational signature analysis amounts to finding a finite set of prototypical mutation patterns and expressing each sample as a sum of these signatures with different weights reflecting the variable exposures in each sample. Mathematically, this process can be modelled by non-negative matrix factorisation into lower dimensional exposure and signature matrices. TensorSignatures generalises this framework by expressing the (expected value of the) count tensor as a product of an exposure matrix and a signature tensor (Fig. 1b; Methods). Counts are modelled by an overdispersed negative binomial distribution, which is a robust statistical model that also enables to choose the number of signatures with established statistical model selection criteria, such as the Bayesian Information Criterion (BIC) as evidenced by extensive simulations (Supplementary Fig. 1).

The key innovation of TensorSignatures is that the signature tensor itself has a lower dimensional structure, reflecting the effects of different genomic features (Fig. 1c). This enables the model to simultaneously learn mutational patterns and their genomic properties by drawing information from the whole dataset, even when the number of combinations of genomic states becomes high (1152). In this parametrisation each signature is represented as a set of $2 \times 2$ strand-specific mutation spectra and

a set of defined genomic activity coefficients, measuring the relative activity of every signature in each state of a given genomic feature. Simulation studies show that this joint inference of mutation spectra and genomic features provides a more accurate inference in comparison to conventional NMF relying on a 96-trinucleotide channel decomposition only and subsequent assessment of signature properties, or post-hoc posterior probability calculations (Supplementary Figs. 2 and 3, Methods).

Furthermore, TensorSignatures incorporates the effect of other variants (MNVs, indels, SVs), which remain unoriented and are expressed as a conventional count matrix, by sharing the same exposure matrix as SNVs. This enables to jointly learn mutational processes across different variant classes more robustly in comparison to approaches which rely on (post-hoc) matching mutational spectra (Supplementary Fig. 4, Methods).

**Most mutational signatures are composed of diverse mutation types and vary across the genome.** To assess the capabilities of TensorSignatures, we analysed the somatic mutational catalogue of the PCAWG cohort comprising 2778 curated whole-genomes from 37 different cancer types containing a total of 48,329,388 SNVs, 384,892 MNVs, 2,813,127 deletions, 1,157,263 insertions and 157,371 SVs. Applying TensorSignatures to the PCAWG dataset and using the conservative BIC (Supplementary Fig. 5) produced 20 tensor signatures (TS) encompassing mutational spectra for SNVs and other mutation types (Fig. 2a), and associated genomic properties (Fig. 2b). Reassuringly, we extracted a number of signatures with SNV spectra highly similar to the well curated catalogue of COSMIC signatures[9,33]. 14/20 signatures were similar to those detected by a de novo analysis of SigProfiler[34], which detected 22 signatures, two of which considered to be sequencing artefacts (Supplementary Fig. 6).

Interestingly, our analysis revealed a series of signatures that have similar SNV spectra in common, but differ with regard to their genomic properties or mutational composition. These signature splits indicate how mutational processes change across the genome and will be discussed in further detail below. In the following, we refer to signatures via their predominant mutation pattern and associated genomic properties. Of the 20 signatures, 4 were observed in nearly every cancer type (Fig. 2c): TS01, characterised by C > T mutations in a CpG context, most likely due to spontaneous deamination of 5meC, similar to COSMIC SBS1, TS02 of unknown aetiology, and two signatures with relatively uniform base substitution spectra, TS03 (unknown/quiet chromatin), and TS04 (unknown/active chromatin), which loosely correspond to SBS40 and SBS5.

While the most prevalent mutations are single base substitutions, there are 16/20 signatures with measurable contributions from other mutation types (>1%; Fig. 2b). The most notable cases are TS15, which is similar to a compound of COSMIC signatures SBS6/15/26 + ID1/2 and characterised by C > T transversions in a GCN context and frequent mononucleotide repeat indels indicative of mismatch repair deficiency (MMRD). Similarly, TS16, likely to reflect concurrent MMRD and *POLE* exonuclease deficiency, exhibits large probabilities for deletions and a base substitution pattern similar to SBS14. Large proportions of SVs (~25%) were found in TS11, which reflects SV-associated APOBEC mutagenesis caused by double strand break repair with a base substitution spectrum similar to SBS2/13. Furthermore, TS19 apparently reflects a pattern of homologous recombination deficiency (HRD), characterised by a relatively uniform base substitution pattern similar to SBS3, but a high frequency of SVs, in particular tandem duplications (Supplementary Note 1 and Fig. 93).

9/20 signatures displayed a measurable propensity to generate clustered mutations (>0.1%; Fig. 2b). The proportions of clustered

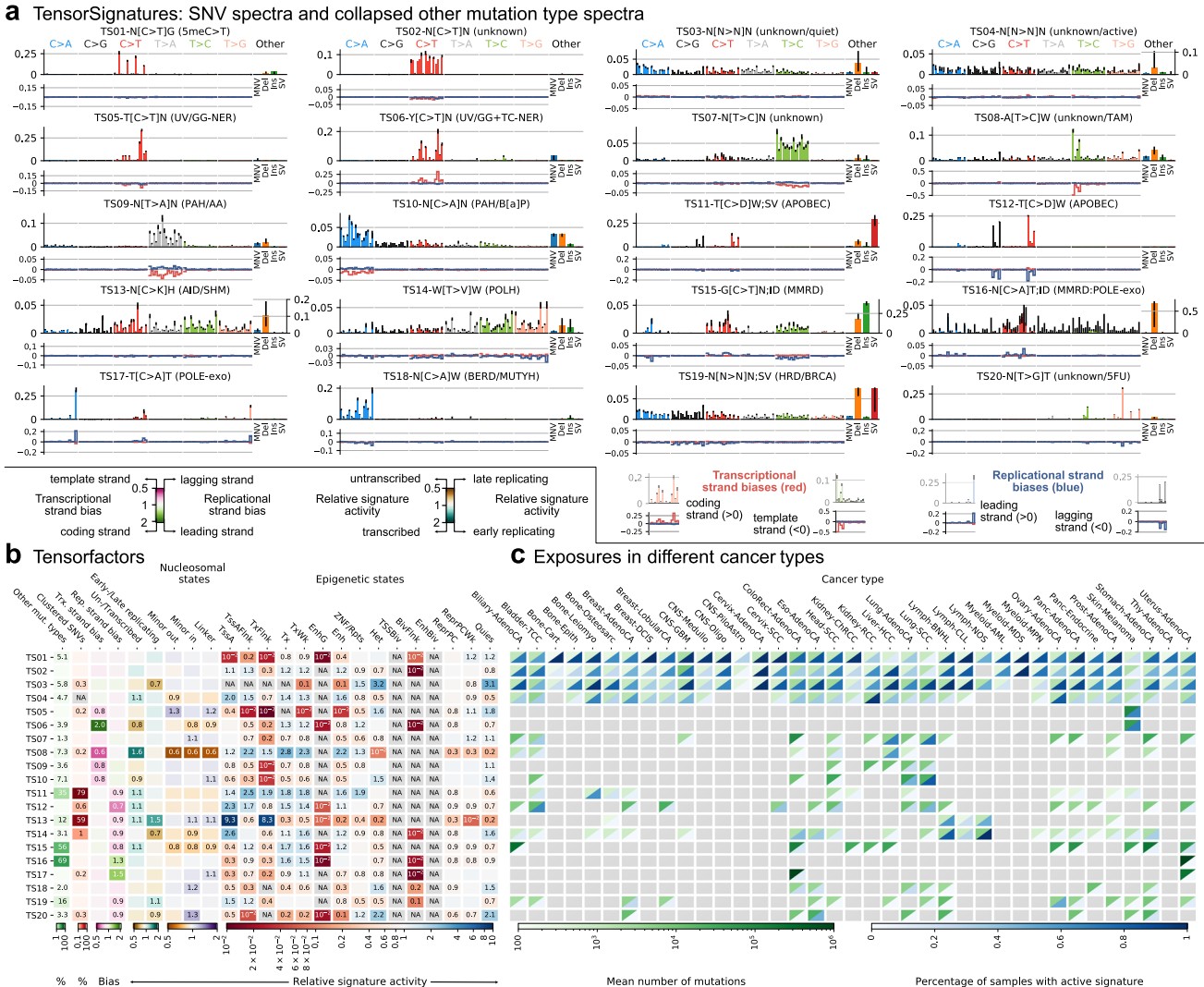

**Fig. 2 Applying TensorSignatures on 2778 whole genomes from the ICCG PCAWG consortium revealed 20 tensor signatures and their genomic properties. a** Upper panels depict SNV spectra, and a summarised representation of associated other mutation types (error bars determine 95% bootstrap confidence intervals). SNV mutations are shown according to the conventional 96 single base substitution classification based on mutation type in a pyrimidine context (blue C > A, black C > G, red C > T, green T > A, grey T > C, salmon T > G) and 5' and 3' flanking bases (in alphabetical order). The panel under each SNV spectrum indicates transcriptional (red), and replicational strand biases (blue) for each mutation type, in which negative deviations indicate a higher probability for template or lagging strand pyrimidine mutations, and positive amplitudes a larger likelihood for coding or lagging strand pyrimidine mutations (and vice versa for purine mutations). **b** Heatmap visualisation of extracted tensor factors describing the genomic properties of each tensor signature. Proportions of other mutation types and clustered SNVs are indicated in percentages. Transcriptional and replicational strand biases indicate shifts in the distribution of pyrimidine mutations on coding/template and leading/lagging strand. Coefficients < 1 (pink) indicate signature enrichment on template or lagging strand DNA, and conversely values > 1 (green), a larger mutational burden on coding or leading strand (a value of 1 indicates no transcriptional or replicational bias). Relative signature activities in transcribed/untranscribed and early/late replicating regions. Coefficients > 1 (turquoise) indicate enrichment in transcribed and early replicating regions, while values < 1 (brown) indicate a stronger activity of the mutational process in untranscribed or late replicating regions. Relative signature activities on nucleosomes and linker regions, and across epigenetic states as defined by consensus ChromHMM states. Scores indicate relative signature activity in comparison to genomic baseline activity. A value of 1 means no increase or decrease of a signature's activity in the particular genomic state, while values > 1 indicate a higher, and values < 1 imply a decreased activity. **c** Signature activity in different cancer types (Exposures). Upper triangles (green) indicate the mean number of mutations contributed by each signature, lower triangles show the percentage of samples with a detectable signal of signature defined as the number of mutations attributed to the signature falling into a signature-specific typical range (Methods). Greyed boxes indicate cancer types for which a signature was not found to contribute meaningfully. Source data are provided as a Source Data file.

mutations produced by each mutational process were highest in signatures associated with APOBEC and activation-induced deaminase (AID) activity: Up to 79% and 0.6% of SNVs attributed to TS11 and TS12, respectively, were clustered, with otherwise indistinguishable base substitution spectra. A similar phenomenon was observed in two signatures reflecting Polη driven somatic

hypermutation (SHM). While both TS13 and TS14 have only mildly diverging base substitution spectra, with TS14 being similar to SBS9, they dramatically differ in the rates at which they generate clustered mutations, which are 59% and 1%, respectively (Fig. 2b).

5/20 signatures exhibit substantial transcriptional strand bias (TSB ≥10%; Fig. 2b). This is strongest in the UV-associated

signature TS06, similar to SBS7b, where the rate of C > T substitutions on the template strand was half of the corresponding value on the coding strand, highly indicative for active TC-NER. In contrast, TS08, similar to SBS16, shows largest activities in liver cancers and preferably produces T > C transitions on template strand DNA. In line with a transcription-coupled role, the activity of TS08 shows a noteworthy elevation in transcribed regions. Both signatures will be discussed in more detail later on.

Analysis of pyrimidine/purine shifts in relation to the direction of replication indicated 9/20 signatures with replication strand biases (RSB ≥10%). In accordance with previous studies, TS12 asserts a higher prevalence of APOBEC-associated C > D mutations, consistent with cytosine deamination, on lagging strand DNA which is thought to be exposed for longer periods as opposed to more processively synthesised leading strand DNA[18–20]. Conversely, TS17, associated with *POLE* exonuclease variants (SBS10a/b), displays a pyrimidine bias towards the leading strand[18] (Fig. 2b). Since DNA polymerase ε performs leading strand synthesis, the strand bias indicates that C > A (G > T) mutations arise on a template C, presumably through C·dT misincorporation[35]. Further examples with replication strand biases include the MMRD-associated signatures TS15 and TS16 discussed above. Of note, the two SHM-associated signatures TS13 and TS14 displayed opposing patterns with respect to their activity in oriented (early) and unoriented (late) replicating regions (Fig. 2b).

Furthermore, all signatures showed signs of differential activity with respect to their genomic features. This was particularly pronounced for epigenetic effects and could be grouped in broadly two classes: Those that are elevated in active (TssA, TssAFlnk, TxFlnk, Tx and TxWk) and depleted in quiescent regions (Het, Quies), and vice versa. This phenomenon includes the two omnipresent signatures with relatively uniform spectra TS03 and TS04, suggesting a mechanism associated with the chromatin state behind their differential manifestation (Fig. 2a). This also applies to two signatures associated with UV exposure, TS05 and TS06, and also two signatures of unknown aetiology, most prominently found in Liver cancers, TS07, similar to SBS12, and TS08, which we will discuss in more detail in subsequent sections. To ensure that these epigenetic associations were not an artefact of the consensus annotation, we matched cancers to their closest Roadmap cell-line(s) (Supplementary Table 1) and performed a TensorSignatures extraction, which yielded highly concordant epigenetic signature activities (Supplementary Fig. 7).

**Validation of TensorSignatures in the HMF cohort**. The aforementioned observations were replicated in a fully independent second cohort of whole genomes from the Hartwig Medical Foundation with 3,824 samples from 31 cancers encompassing 95,531,862 SNVs, 1,628,116 MNVs, 9,228,261 deletions, 5,408,915 insertions and 1,001,433 structural variants[25]. Applying TensorSignatures to this data set produced 27 tensor signatures (Supplementary Fig. 8a). Of these 10 closely resembled (cosine distance < 0.2) signatures of the discovery analysis with closely matching genomic activity coefficients (Fig. 3d, Supplementary Fig. 8b). These include the signatures of spontaneous deamination TS01, the two signatures of UV mutagenesis TS05/06, SV-associated APOBEC mutagenesis TS11, as well as signatures of MMRD TS16, *POLE*[exo] mutations TS17, as well as *MUTYH* deficiency TS18, HRD TS19 and TS20.

A further 7 signatures seemingly constitute splits of tensor signatures from the PCAWG cohort (Fig. 3b). A complex three-way split appeared to occur for TS03 and TS04, which were found in a broad range of cancer types. One of the derivative signatures resembles the mutation spectrum of SBS8 from the COSMIC

catalogue, however without measurable transcriptional strand bias. A second derived signature is similar to SBS39; our analysis reveals replication strand bias for C > G variants and a potentially wider range of cancer types for both signatures. Further, signature TS12, resembling replication associated APOBEC mutagenesis, split into two signatures with base substitution spectra similar to SBS2 (C > T) and SBS13 (C > G), but preserving the strong replication strand bias. Lastly, a split of TS10, likely attributed to mutagens included in tobacco smoke, was observed.

Finally, a set of 10 signatures without close match to those in the PCAWG cohort was found (Fig. 3c). This includes five spectra linked to cancer therapies, illustrating the additional insights on preceding therapies provided by the HMF metastatic cancer cohort. TS21 is characteristic of treatment with the methylating agent temozolomide (SBS11); the observed transcriptional strand bias reflects a higher rate of G > A mutations on the coding strand (equivalent to higher rates of C > T on the template strand), consistent with methyl guanine being removed by TC-NER in the absence of MGMT. TS22 and TS23 have been previously associated with cisplatin (termed E-SBS21 and E-SBS14)[36,37]. While both signatures exhibit mild transcriptional strand biases, only TS23 shows a strong association with MNVs going in line with the propensity of cisplatin/oxaliplatin to form intrastrand DNA adducts (Supplementary Fig. 8c). TS24 displays the characteristics of treatment with 5-FU, which inhibits thymine synthesis and has been proposed to be mutagenic via genomic fluorouracil incorporation[37]. TS28, with similarity to SBS41, was only found in two samples, possibly due to treatment with the experimental drug SYD985, which consists of a duocarmycin-based HER2-targeting antibody-drug conjugate[25].

Further, TensorSignatures detected a signature of colibactin, TS25, which has been previously characterised[36,38]. TS25 displays contributions of MNVs and short indels, activity in active genomic regions and concomitant transcriptional strand bias of T > C mutations (Fig. 3c, d). TS26's indels and similarity to SBS15 suggests an association with MMRD; TS27 has an unknown aetiology and displays strong replicational strand bias. The large proportion of structural variants and the flat SNV spectrum of TS29 may represent non-specific mutagenesis at SVs. TS30 was found in lymphoid and other cancers and had a high proportion of clustered mutations, similar, but not identical to TS14 (Fig. 3d).

Taken together, the TensorSignatures analysis of the PCAWG and HMF cohorts revealed that mutational signatures are composed of diverse mutation types and vary extensively across the genome. In the following, we explore some of the genomic activity patterns and properties of selected TensorSignatures in further detail. An emerging feature was differential mutagenesis in active and quiet areas of the genome.

**The spectrum of UV mutagenesis changes from closed to open chromatin, reflecting GG- and TC-NER**. Two signatures, TS05 and TS06, were exclusively occurring in Skin cancers of both cohorts and displayed almost perfect correlation (Spearman $R^2 = 0.98$, Supplementary Fig. 9a) of attributed mutations, strongly suggesting UV mutagenesis as their common cause. Both signatures share a very similar SNV spectrum, only differing in the relative extent of C[C > T]N and T[C > T]N mutations, which is more balanced in TS06 (Fig. 2a). However, they strongly diverge in their activities for epigenetic contexts and transcriptional strand biases: TS05 is enriched in quiescent regions, and shows no transcriptional strand bias, while the opposite is true for TS06, which is mostly operating in active chromatin (Fig. 2b). Of note, the spectra of these signatures closely resemble that of COSMIC SBS7a and SBS7b, which have been suggested to be linked to different classes of UV

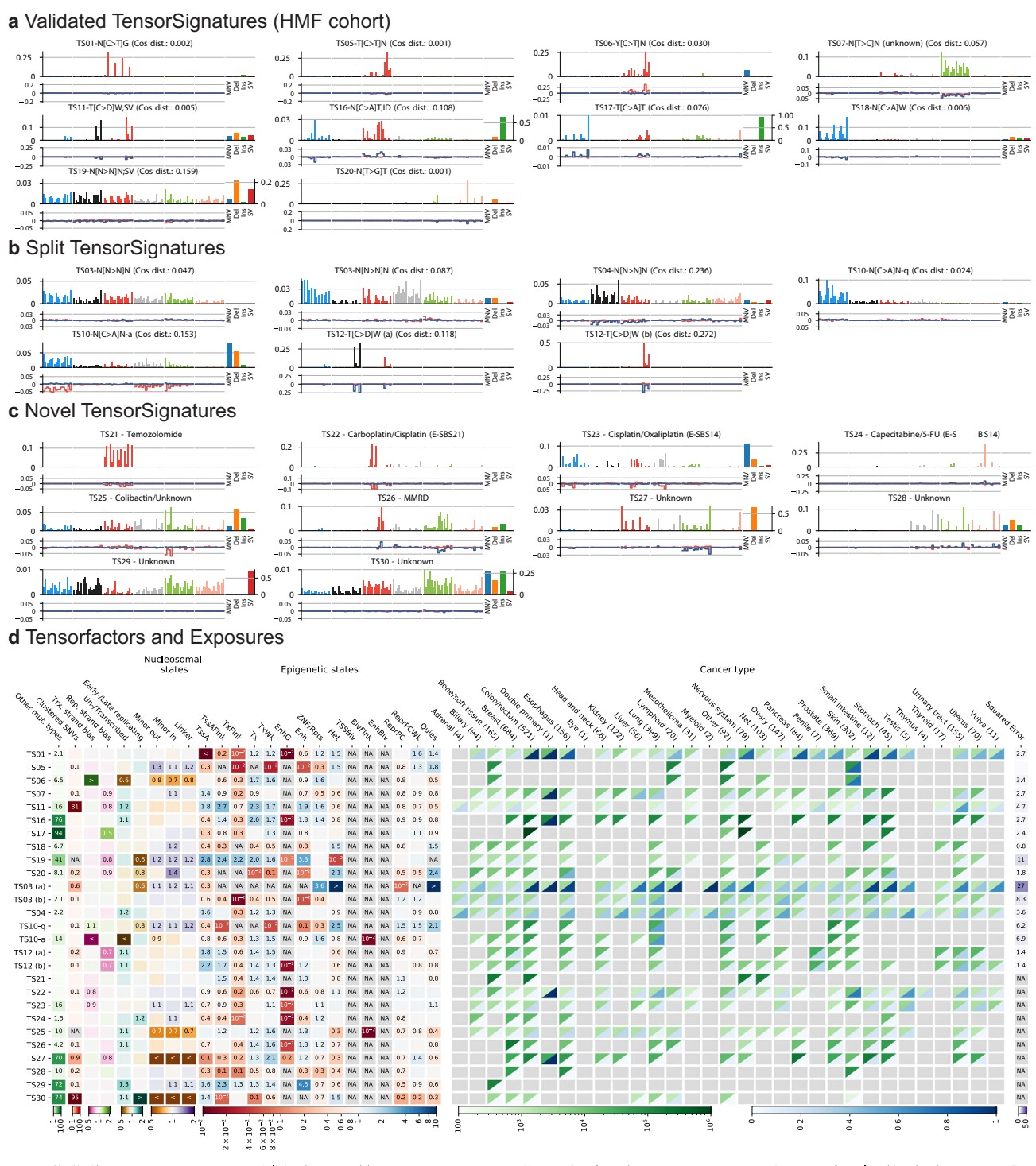

**Fig. 3 Tensor signatures of the HMF cohort. a** Validated tensor signatures with high similarity (indicated as cosine distance) to the mutational processes extracted in our discovery analysis using PCAWG data. **b** TensorSignatures splits that seemingly represent derivatives of tensor signature TS03, TS04, TS10 and TS12. **c** Tensor signatures of the HMF cohort. **d** Extracted tensor factors, exposures and summed squared errors of tensor factors from the discovery and validation analysis. Source data are provided as a Source Data file.

damage[39]. However, as our genomically informed TensorSignatures inference and further analysis show, the cause for the signature divergence may be found in the epigenetic context, which seemingly not only determines mutation rates, but also the resulting mutational spectra.

A characteristic difference between the two signatures is the presence of a strong transcriptional strand bias in signature TS06,

which is almost entirely absent in signature TS05 (Fig. 4a). To verify that this signature inference is correct, and the observed bias and spectra are genuinely reflecting the differences between active and quiescent chromatin, we pooled C > T variants from Skin-Melanoma samples which revealed that the data closely resembled predicted spectra (Fig. 4b). In addition, quiescent chromatin also displays a predominant T[C > T]N substitution

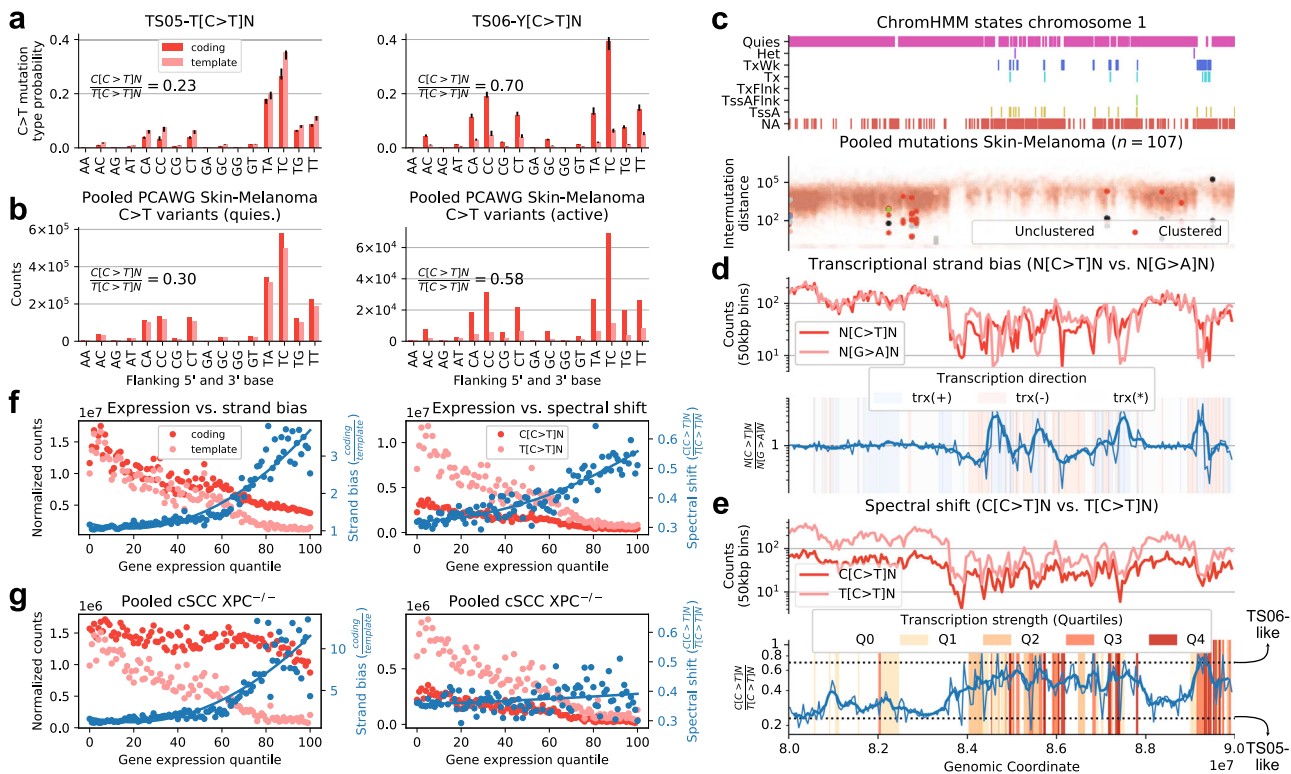

**Fig. 4 The spectrum of UV mutagenesis changes from open to closed chromatin. a** C > T mutation probabilities of TensorSignatures TS05 and TS06 for coding and template strand DNA (error bars determine 95% bootstrap confidence intervals). **b** Pooled PCAWG Skin-Melanoma C > T variant counts from coding and template strand DNA in epigenetically active (TssA, TssAFlnk, TxFlnk, Tx and TxWk, right) and quiescent regions (Het and Quies, left). **c** Consensus ChromHMM states from a representative 10 Mbp region on chromosome 1, and the corresponding mutational density of pooled Skin-Melanoma samples. **d** N[C > T]N and N[G > A]N counts in 50 kbp bins, and their respective ratios (thin blue line: ratio; thick blue line: rolling average over 5 consecutive bins) illustrate the transcriptional strand bias of C > T mutations in quiescent and active regions of the genome. **e** Relationship between expression strength and the spectral shift of C > T mutations in terms of binned C > T variant counts in TpC and CpC context and their respective ratios (thin blue line) as well as a rolling average (thick blue line). **f** Gene expression strength vs. transcriptional strand bias (measured by the ratio normalised C > T variants in Skin-Melanoma on coding and template strand), and gene expression strength vs. C[C > T]/T[C > T] spectral shift (indicated as the ratio of normalise C > T mutations in 5'C and 5'T context). **g** Transcriptional strand bias and C[C > T]/T[C > T] spectral shift in GG-NER deficient $XPC^{-/-}$ cSCC genomes. Blue curves: quadratic fit. Source data are provided as a Source Data file.

spectrum (5'C/5'T = 0.3), while the spectrum in active chromatin is closer to Y[C > T]N (5'C/5'T = 0.58), as predicted by the signature inference (Fig. 4a). This difference does not appear to be related to the genomic composition, and holds true even when adjusting for the heptanucleotide context (Supplementary Fig. 9b).

To illustrate how the mutation spectrum changes dynamically along the genome in response to the epigenetic context, we selected a representative 10 Mbp region from chromosome 1 comprising a quiescent and active genomic region as judged by consensus ChromHMM states, and the varying mutational density from pooled Skin-Melanoma samples (Fig. 4c). As expected, actively transcribed regions display a strong transcriptional strand bias (Fig. 4d). Further, this change is also accompanied by a change of the mutation spectrum from a T[C > T]N pattern to a Y[C > T]N pattern with the ratios indicated by our TensorSignatures inference (Fig. 4e).

These observations are further corroborated by RNA-seq data available for a subset of samples (n = 11): The transcriptional strand bias is most pronounced in expression percentiles greater than 50 leading to an increased ratio of coding to template strand mutations (Fig. 4f). Again, the decline is accompanied by a shift in the mutation spectrum: While both C[C > T]N and T[C > T]N variant counts decline steadily as gene expression increases, the reduction of C[C > T]N mutations is larger in comparison to

T[C > T]N mutations, which manifests as an increasing C[C > T]N and T[C > T]N ratio, reaching a ratio of approximately 0.5 in the highest expression quantiles (Fig. 4f).

The diverging activity in relation to the chromatin state suggests an underlying differential repair activity. Global genome nucleotide excision repair (GG-NER) clears the vast majority of UV-lesions in quiescent and active regions of the genome and is triggered by different damage-sensing proteins. Conversely, TC-NER is activated by template strand DNA lesions of actively transcribed genes. As TS05 is found in quiescent parts of the genome, it appears likely that it reflects the mutation spectrum of UV damage as repaired by GG-NER. Based on the activity of TS06 in actively transcribed regions and its transcriptional strand bias, it seemingly reflects the effects of a combination of GG- and TC-NER, which are both operating in active chromatin. This joint activity also explains the fact that the spectrum of TS06 is found on both template and coding strands.

This attribution is further supported by data from n = 13 cutaneous squamous cell carcinomas (cSCCs) of n = 5 patients with Xeroderma Pigmentosum, group C, who are deficient of GG-NER and n = 8 sporadic cases which are GG-NER proficient[40]. *XPC*/GG-NER deficiency leads to an absence of TS05 in quiescent chromatin and to a mutation spectrum that is nearly identical in active and quiescent regions of the genome (Supplementary Fig. 9c). Furthermore, the UV mutation

spectrum of *XPC*/GG-NER deficiency, which is thought to be compensated by TC-NER, differs from that of TS06, reinforcing the notion that TS06 is a joint product of GG- and TC-NER. This is further supported by the observation that *XPC*/GG-NER deficiency leads to a near constant coding strand mutation rate, independent of transcription strength[40] (Fig. 4g), indicating that the transcriptional dependence of coding strand mutations in GG-NER proficient melanomas and cSCCs is due to transcriptionally facilitated GG-NER.

While the activity patterns of TS05 and TS06 appear to be well aligned with GG-NER and GG/TC-NER, these observations, however, do not explain the observed differences in mutation spectra. The fact that the rates of C[C > T]N and T[C > T]N mutations change between active and quiescent chromatin—and the fact the these differences vanish under *XPC*/GG-NER deficiency—suggests that DNA damage recognition of CC and TC cyclobutane pyrimidine dimers by GG-NER differs between active and quiescent chromatin, with relatively lower efficiency of TC repair in quiescent genomic regions, as evidenced by TS05.

**A differential mutation spectrum of tobacco-associated mutations in regions with TC-NER.** A similar split of an exogenous mutational signature into quiet and active chromatin was observed in lung cancers of the HMF cohort where TS10 splits into two signatures, HMF TS10-q, which shows largest activity in heterochromatin, while HMF TS10-a is enriched in actively transcribed regions, and exhibits a strong transcriptional strand bias with lower rates of C > A changes on the coding strand, equivalent to G > T transversions on the template strand (Figs. 3b, d and 5a). This strand bias has been attributed to TC-NER removing benzo[a]pyrene derived adducts on guanines from the template strand[41].

The emergence of two mutational signatures indicates that this repair process also changes the mutation spectrum. The suggested split is also evident in pooled mutations from HMF lung cancers in quiescent and active genomic regions, respectively, revealing that predicted spectra coincide with corresponding tensor signatures HMF TS 10-q and TS10-a (Fig. 5b). The C > A (G > T) mutation spectrum observed in quiescent regions, extracted by HMF TS10-q, displays highest rates of mutations in a CCN (NGG) context (Fig. 5a). Interestingly, the same pattern is also observed in actively transcribed regions for C > A on the template strand, equivalent to G > T mutations on the coding strand. This is in contrast to the C > A coding strand pattern, and HMF TS10-a, for which this difference is largely eroded. These observations reflect how TC-NER removes genotoxic guanine adducts from the template strand, which leads to lower mutation rates and also a more homogeneous base context of G > T mutations. The differentential mutation spectrum indicates that either the efficiency of TC-NER—or the mutagenicity of residual genomic alterations—differs depending on the base context, analogous to observations in UV-induced mutagenesis. The result being that the mutation types and rates caused by tobacco-associated carcinogens differ between coding and template strand in transcribed regions and also to different mutation spectra in quiescent and active genomic regions.

**Transcription-associated mutagenesis is common in highly transcribed genes.** A third split of mutational signatures between active and quiet regions was observed in Liver and other cancer types (Fig. 2b, c), driven by differential activity of TS07 and TS08, which closely resemble COSMIC signatures SBS12 and SBS16, respectively. In line with previous findings[18,26], there was a strong transcriptional bias of TS08, introducing 1.6 × more T > C variants on the template strand (Fig. 2b). While both signatures are most frequently found in Liver cancers, where they are strongly correlated ($R^2 = 0.68$, Supplementary Fig. 10a), they are also observed in a range of other cancers, indicating that they are reflecting endogenous mutagenic processes.

The most prominent difference between these signatures is the depletion of mutation types in 5'-B context on coding strand DNA in TS08 (Fig. 5c; B = C, G, or T). Signature TS08 displays a strong transcriptional strand bias, as previously noted for SBS16[26], and is confirmed here by a direct investigation of variant counts (Fig. 5d). A further defining feature of TS08 are indels ≥2 bp (Fig. 2a, Supplementary Note 1 Fig. 38), which were reported to frequently occur in highly expressed lineage-specific genes in cancer[12], consistent with experimental data of transcription-replication collisions[42]. In line with this, mutation rates showed a dynamic relation to transcriptional strength (Fig. 5e). Normalised counts of T > C mutations on coding and template strand initially decline for low transcription. Yet this trend only continues on the coding strand for transcription quantiles (>50), but reverses on the template strand, producing more N[T > C]N, and most commonly A[T > C]N, mutations the higher the transcription, in line with previous reports of TAM[18].

While this effect is most common in Liver-HCC samples, where it has been described in detail, it has been observed that SBS5, one of three broadly active signatures, displays signs of potential contamination by SBS16/TS08 in the absence of further intra-genomic stratification. Accordingly, a genomically informed analysis by TensorSignatures also discovers this signature in highly transcribed genes of Head-SCC, Stomach-AdenoCa and Biliary-AdenoCa (Fig. 5f, Supplementary Fig. 10b), showing that A[T > C]W TAM and N[T > C]N mutagenesis in heterochromatic regions occur in a broad range of cancers.

**Replication- and DSBR-driven mutagenesis by APOBEC3A and 3B.** APOBEC mutagenesis has been previously studied in detail in cancer genomes[1,30,43], revealing and localisation to double strand breaks (kataegis)[1] and replication strands bias[18–20]. Further investigations in experimental systems and cancer genomes discovered an extended base context characteristic of the enzymes APOBEC3A and APOBEC3B, respectively revealing a high contribution of APOBEC3A[27] across the genomic background and elevated contributions of APOBEC3B to SV-associated mutation clusters[28,44]. Still, these facets are not part of current reference catalogues; instead SBS2 and SBS13 discriminate the effects base excision repair operating downstream of APOBEC induced uracil, producing C > G (SBS2) and C > T substitutions (SBS13), respectively. The unified approach of TensorSignatures may help consolidate some of the observed facets of APOBEC mutagenesis. For example, it has not been studied whether the sparse set of clusters attributed to APOBEC3A display a replication strand bias, which would confirm them to arise during (lagging strand) replication.

TensorSignatures TS11 and TS12 share a common base substitution spectrum, but differ greatly with regard to their genomic properties: While TS12 is dominated by SNVs (99%) with strong replicational strand bias, SNVs in TS11 make up only 64% of the overall spectrum and are highly clustered. The rest of the spectrum is mostly dominated by structural variants (Figs. 2a and 6a, Supplementary Note 1 Fig. 53). Reassuringly, the difference in clustering propensity and replication strand bias are also directly apparent in the mutation spectra and rainfall plots of samples with high contributions of either signature (Fig. 6b, c). SV proximal and distal clustered variants do not display a replicational strand bias, indicating that both arise in a DSB driven manner, the latter presumably during successful DSBR, which did not manifest as SV (Supplementary Fig. 11a).

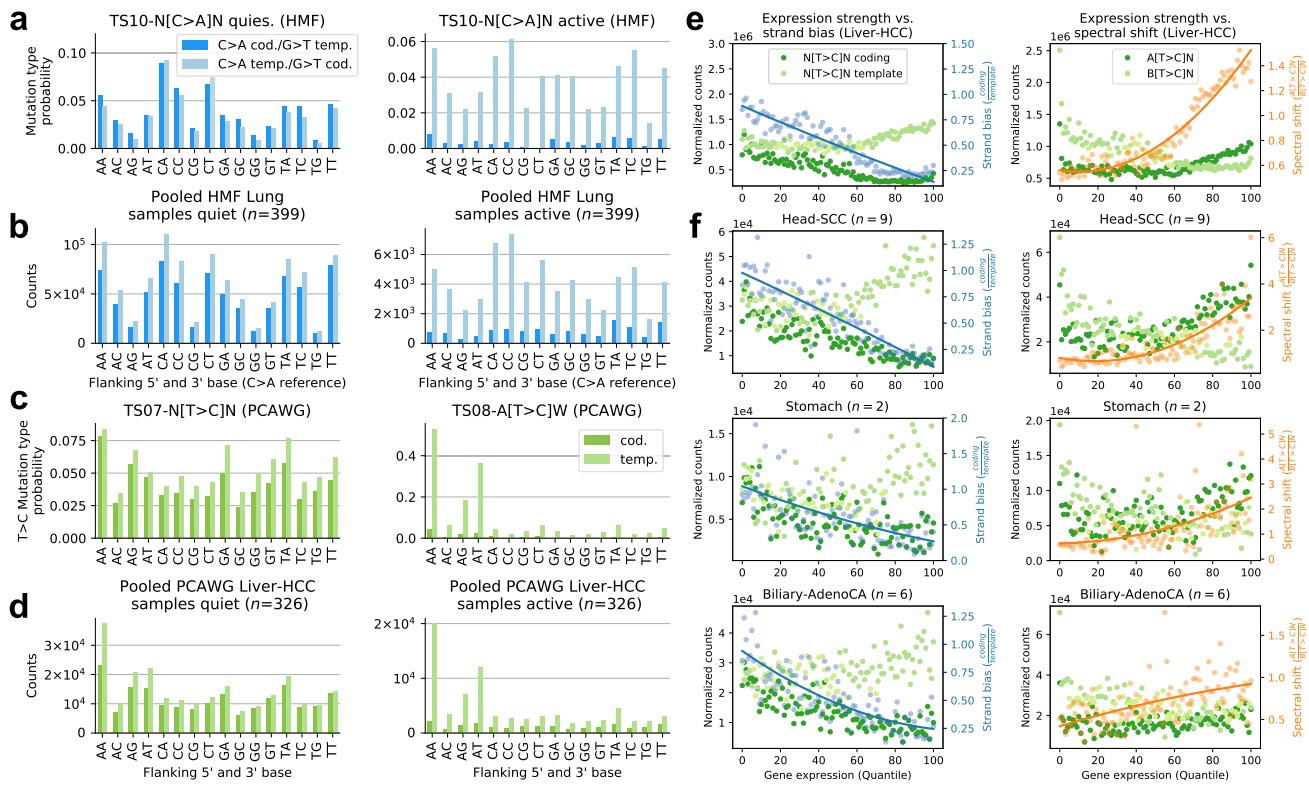

**Fig. 5 Genomically dependent T > C mutagenesis in Liver-HCC and other cancer types. a** C > A mutation type probabilities of HMF TS10-q and TS10-a for coding and template strand DNA. **b** Pooled HMF lung samples C > A variant counts from coding and template strand DNA in quiescent (Het and Quies, left) and epigenetically active regions (TssA, TssAFlnk, TxFlnk, Tx and TxWk, right). **c** T > C mutation type probabilities of PCAWG TS07 and TS08 for coding and template strand DNA. **d** Pooled PCAWG Liver-HCC T > C variant counts for coding and template strand DNA in epigenetically active and quiescent regions. **e**, **f** Transcriptional strand bias and A[T > C]/B[T > C] spectral shift in samples from different cancers with TS07 and TS08 contributions. Lines correspond to quadratic fits. Source data are provided as a Source Data file.

However, clusters at SVs tend to be larger (Median 717 vs. 490 bp) and tend to have more mutations per cluster (Median 5 vs 4 variants; Fig. 6d), suggesting a higher processivity of APOBEC mutagenesis during DSBR. To further the link of these signatures to APOBEC3A and APOBEC3B mutagenesis, we studied the extended motifs YT[C > T]A and RT[C > T]A, which suggest a higher prevalence of purines (R) over pyrimidines (Y) at the -2 5' nucleotide[27]. Indeed, clustered TS12 mutations comprise only a low fraction of purines, indicative of APOBEC3B, while this proportion increases to approximately 50% in TS11 samples, consistent with the reported motif of APOBEC3A (Fig. 6e, Supplementary Fig. 11b).

Finally, we note that, using data from the HMF cohort, TS12, was further split into C > T and C > G akin to mutational signatures SBS2 and 13 from the COSMIC catalogue (Fig. 3b). These two signatures were attributed to differential activity of OGG/UNG-driven base excision repair of uracil created by APOBEC-induced cytosine deamination[45]. The observation that this split occurs for TS12 rather than TS11, suggests that repair of replication-driven APOBEC3A mutagenesis, may be subject to higher variation in downstream BER than DSBR-driven APO-BEC3B mutagenesis.

**Targeted somatic hypermutation at TSS and dispersed clustered translesion synthesis.** Two other TensorSignatures produced substantial amounts of clustered variants with, but different epigenomic localisation. TS13 showed largest activities in lymphoid cancers and produced 60% clustered variants (Fig. 2b). The SNV spectrum resembles the c-AID signature

reported previously[7], suggesting an association with activation-induced cytidine deaminases (AID), which initiates somatic hypermutation in immunoglobulin genes of germinal centre B cells[46,47].

TensorSignatures analysis finds that TS13 activity is 9 × and 8× enriched at active transcription start sites (TssA) and flanking transcription sites (TxFlnk, Fig. 2b), respectively. To illustrate this, we pooled single base substitutions from Lymph-BHNL samples and identified mutational hotspots by counting mutations in 10 kbp bins (Fig. 7a, b), which revealed that clustered mutations often fell accurately into TssA regions (Fig. 7c). The aggregated clustered mutation spectrum in TssA/TxFlnk regions across lymphoid neoplasms (Lymph-BNHL/CLL/NOS, $n = 202$) indeed showed high similarity to TS13, possibly with an even more pronounced rate of C > K (K = G or T) variants similar to SBS84[9] (Fig. 7d). Conversely, the clustered mutational spectrum from all other epigenetic regions was characterised by a larger proportion of T > C and T > G mutations, similar to TS14, which only produces about 1% clustered mutations and closely resembles SBS9, attributed to Polη-driven translesion synthesis (TLS) during somatic hypermutation[45].

While TS13 and TS14 are strongly correlated ($R^2 = 0.88$, Supplementary Fig. 12), the diverging localisation pattern and SNV spectrum, characterised by higher rates of C > K mutations in TS13, indicates that a related, but different mutational process drives TSS hypermutation, seemingly linked to AID. The differential mechanism behind TS13 also manifests as longer clusters (Median: 1068 vs. 183 bp), which contain more variants per cluster (Median: 8 vs. 3 mutations) in comparison to TS14 (Fig. 7e). As a further distinction, the weakly clustered TLS

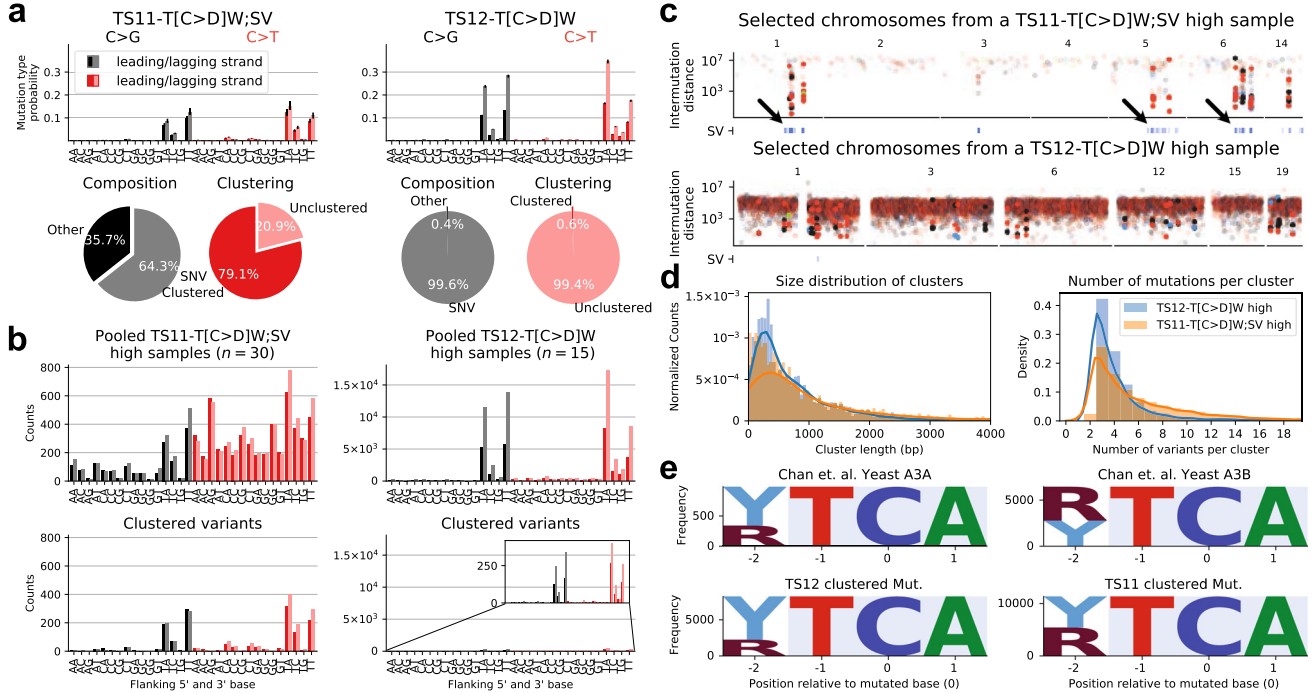

**Fig. 6 Double-strand break and replication induced APOBEC mutagenesis. a** C > G and C > T spectra of TS11 and TS12 for leading and lagging strand DNA (error bars determine 95% bootstrap confidence intervals). Pie charts underneath indicate percentages of clustered mutations and the contribution of other mutation types in TS11 and TS12. **b** Observed unclustered (top) and clustered variants (bottom) in TS11 and TS12 high samples (TS11 and TS12 contributions > 10% and 70% respectively). **c** Rainfall plots with SV annotations from a typical sample with high TS11 (top) and TS12 contributions (bottom). **d** Size distribution of mutation clusters (consecutive clustered mutations), and the distribution of number of variants per mutation cluster in TS11 and TS12 high samples respectively. Curves depict corresponding kernel density estimates. **e** Motif logo plots of the tetranucleotide context at mutated TCA sites in yeast cells exposed to APOBEC3A/3B mutagenesis respectively[27], and similar motif logo plots extracted at clustered mutations from samples with high TS11 or TS12 exposures. Source data are provided as a Source Data file.

signature TS14 can be found in more than 15 cancer types, suggesting a broad involvement of this mutagenic process in resolving endogenous and exogenous DNA alterations[28]. Finally, a third mutational signature of somatic hypermutation, TS30, was found in lymphoid and other cancers of the HMF cohort. This signature displayed a large proportion of clustered mutations and an enrichment in early replicating regions similar to TS13, combined with an SNV spectrum that was closer to TS14 (Cosine distance $d = 0.13$ vs. 0.25), suggesting that TS30 may represent a combination of TS13 and TS14.

## Discussion

We presented TensorSignatures, a framework for learning mutational signatures jointly from their mutation spectra and genomic properties to better understand the underlying mutational processes. We illustrated the capabilities of this algorithm by presenting a set of 20 mutational signatures extracted from 2778 cancer genomes of the PCWAG consortium, and validated our analysis on additional 3824 metastatic samples from the HMF cohort. The analysis demonstrated that the majority of mutational signatures comprised different variant types, and that no single mutational signature acted uniformly along the genome. Measuring how mutational spectra are influenced by their associated genomic features sheds light on the mechanisms underlying mutagenesis, as demonstrated by multiple previous investigations[18–20]. However, as such calculations have been carried out after defining mutational signatures, they cannot detect more subtle signature changes associated with genomic features and struggle to localise very similar mutational signatures. A joint inference also helps to dissect mutational processes in situations where mutation spectra are very similar, such

that genomic associations cannot be unambiguously attributed based on the mutation spectrum alone.

In comparison to comparable tools for mutational signature analysis, TensorSignatures is currently not optimised to maximise the number of extracted signatures, but rather to characterise the properties of recurrent mutational processes. Compared to the curated COSMIC catalogue of mutational signatures, the automated analysis with TensorSignatures currently misses the signatures of of Aflatoxin (SBS24, $n = 2$ samples with relative exposure >5%), platinum therapy (SBS31, $n = 2$; SBS35, $n = 11$) and a signature characteristic of base excision deficiency by *NTHL1* defects (SBS30, $n = 45$), which in aggregate are estimated to contribute 0.2% of single base substitutions in the PCAWG cohort. One reason for this discrepancy is that the COSMIC catalogue is informed by an additional 1865 whole genome samples from other sources, which provide stronger evidence for rarely recurring mutational signatures. In order to maximise the yield of novel signatures—a somewhat different objective to characterising genomic properties of recurrent processes—a more bespoke analysis recognising each cancer type and its preferred set of mutational signatures as well as a careful assessment of potential sequencing and analysis artefacts is necessary.

Nevertheless, the number of signatures in our discovery analysis closely matched the decomposition rank suggested by SigProfiler (default settings, v1.0.17), indicating that Tensor-Signatures is similarly sensitive, while providing the benefit of characterising extracted mutational processes with respect to a multitude of genomic variables. The observed discrepancy of approximately 30% between methods highlights some of the challenges in mutational signature analysis, which needs to discriminate between genuine biological signal and noise. As our

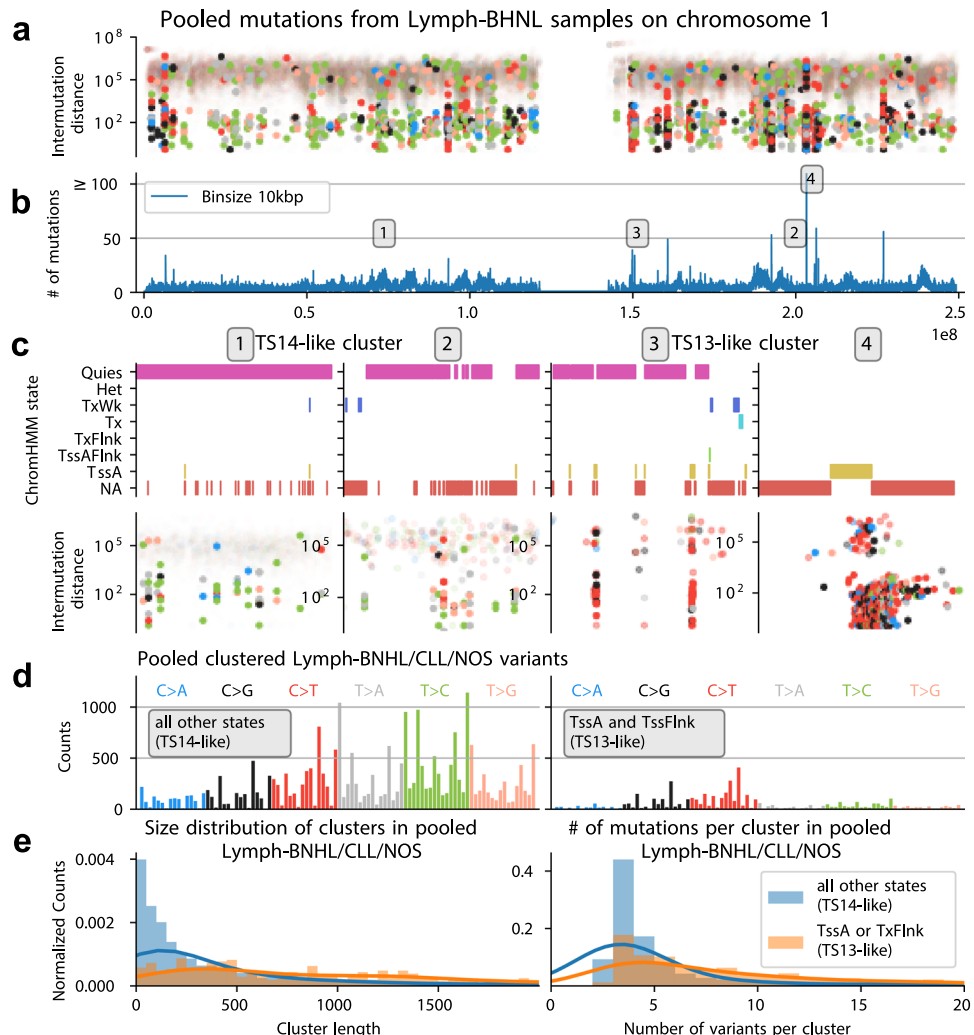

**Fig. 7 Identification of a highly clustered mutational signature at active TSS. a** Rainfall plot of pooled variants from Lymph-BHNL samples on chromosome 1 (highlighted dots indicate clustered mutations). **b** Binned (10 kb) SNV counts of chromosome 1. Numbers 1-4 indicate mutation hotspots. **c** Consensus ChromHMM states and rainfall plots at mutation hotspots. **d** Pooled clustered variants from PCAWG Lymph-BHNL/CLL/NOS samples from TssA or TxFlnk (TS13-like), and all other epigenetic states (TS14-like). **e** Size distribution of mutation clusters (consecutive clustered mutations), and the distribution of number of variants per mutation cluster in TS13 and TS14 high samples respectively. Source data are provided as a Source Data file.

simulations showed, using more types of mutations and genomic features may increase the accuracy of extracting signatures and of measuring their local activities. Further refinements may include to model a preferred activity of a particular signature in a given tissue type; including such preference may help better ascertain the sets of signatures found in a particular genome. Also, incorporating more genomic features, potentially so in quantitative ways, and ideally matching these annotations to each tumour type are likely to increase the power of the analysis. TensorSignatures appears to be well suited for such refined and future analyses as tumour specific genomic annotations are likely to be assembled over the next years. Accompanying these anticipated developments, further simulation tools cognisant of various genomic features have to be developed, as current frameworks do not recognise the genomic distribution of point mutations[9].

Epigenetic annotations, for example, currently exist only for a subset of cancer types—and it may even be that individual cancer subtypes derive from distinct cells of origin with unique epigenomic characteristics. To facilitate a pan-cancer analysis, TensorSignatures uses, similar to previous studies[18,23], consensus annotations comprising those genomic regions found to vary only lowly between different cell types. This reduces the number of

annotated variants by approximately 30% as many parts of the genome are annotated as 'variable' (an extra state introduced by TensorSignatures). An analysis based on partially matched tissues showed that this approach is likely to underestimate the effect of genomic factors on mutagenesis and that the reduction of signal makes the analysis of rare elements, such as enhancers, noisier (Supplementary Fig. 7). As more and more tissues are being genomically profiled, we expect that further tissue-specific annotations—and also entirely new genomic features—will emerge, which will produce more accurate and novel insights into the determinants of mutagenesis.

Studying the signatures discovered in the PCAWG and HMF data sets revealed that the SNV spectra of TS05 and TS06 show high similarity to signatures SBS7a and SBS7b of the COSMIC catalogue of mutational signatures. Due to the high similarity of the mutational spectra, it is difficult to unambiguously attribute individual mutations to these signatures and measure their genomic activity and transcriptional strand biases based on the mutation spectra alone. TensorSignature analysis reveals that the two processes are strongly differing with respect to their epigenetic context and transcriptional strand bias pointing towards differentially active GG-NER to be the underlying cause of the

regional signature, which is confirmed by analysing cSCCs from GG-NER deficient *XPC* patients.

A similar change of the mutation spectrum was observed in Liver-HCC and other cancer types, reflected by the diverging activity of TS07 and TS08. The activity of TS08 is most prominent in highly transcribed genes, indicative of transcription-associated mutagenesis[12,18]. TensorSignatures unifies the overarching mutational spectrum of this process and sheds light on its genomic determinants. Furthermore, its ability to detect mutational signatures in specific genomic regions also increases the sensitivity to detect signature activity, which may only contribute low levels of mutation at a genome wide scale. Here, we find TS08 also in Bladder-TCC, ColoRectal-AdenoCa, Lung-AdenoCa, Prostate-AdenoCa and Stomach-AdenoCa in addition to Billiary-AdenoCa, Head-SCC, and Liver-HCC, where it has been previously found[9].

TensorSignatures' capability to detect signatures with a confined regional context was also highlighted by detecting a highly localised signature associated with AID, TS13, which specifically manifests in and around transcription start sites in lymphoid neoplasms[7]. This signature has a base substitution spectrum similar to TS14 (SBS9), which does not display the tight localisation to TSS and is found in a range of cancer types, likely reflecting Polη-driven TLS during replication.

Inclusion of other mutation types led to the discovery of two APOBEC-associated signatures representative for mutagenesis during replication and at DSBRs, which differ with regard to their replicational strand bias and clustering propensity and are likely to reflect differential activity of APOBEC3A and 3B. While such associations have been reported previously, these two processes are currently not reported as distinct mutational signatures due to their identical base substitution spectrum. TensorSignatures shows that such a distinction can be formally achieved by a joint analysis of overarching mutation spectra and genomic features and shed light on the exact properties and activity of either process.

Taken together, this analysis maps out the regional activity of mutational processes across the genome and pinpoints their various genomic determinants. As mutational signature analysis is an essential element of the cancer genome analysis toolkit, TensorSignatures will help make the growing catalogues of mutational signatures more insightful by highlighting mutagenic mechanisms, or hypotheses thereof, to be investigated in greater depth.

## Methods

**Data acquisition**. We performed the primary analysis on WGS data of primary tumours of the PCAWG dataset, and used WGS data of metastatic tumours from the HMF[25] cohort for the validation analysis. Additionally, we analysed WGS data from XPC deficent cSCCs[48]. We obtained GENCODE v19 definitions and Repli-seq data for GM12818, K564, Hela, Huvec and Hepg2 cell lines from the ENCODE consortium[49,50] to annotate SNVs with transcription and replication orientation. Further, we downloaded 15-state ChromHMM annotations[32] as well as nucleosome dyad positions from MNase cut efficiency experiments[23] to assign epigenetic and nucleosomal annotations to SNV data.

## Count tensor

*Transcription*. To assign single base substitutions to template and coding strand, we partitioned the genome by transcription directionality (trx(+)/trx(−)) using GENCODE v19 definitions. Nucleic acids can only be synthesised in $5' \rightarrow 3'$ direction implying that template and coding strand of trx(−) genes are $5' \rightarrow 3'$ and $3' \rightarrow 5'$ oriented, and vice versa for trx(+) genes. Since mutations are called on the + strand of the reference genome, and representing single base substitutions in a pyrimidine base context, we can unambiguously determine whether the pyrimidine of the mutated Watson-Crick base pair was on the coding or template strand. For example, a G > A substitution in a trx (−) gene corresponds to a coding strand C > T mutation, because the transcription directionality dictates that the mutated G sits on the template strand. Splitting all SNVs in this fashion requires us to introduce

an additional dimension of size three (coding, template and unknown strand) to the count matrix ($\mathbf{C}^{SNV} \in \mathbb{N}_0^{3 \times p \times n}$ where $p = 96$ and $n$ is the number of samples).

*Replication*. To assign single base substitutions to leading and lagging strand, we leveraged Repli-seq data from the ENCODE consortium[49,50], which map the sequences of nascent DNA replication strands throughout the whole genome during each cell cycle phase. Repli-seq profiles relate genomic coordinates to replication timing (early and late), where local maxima (peaks) and minima (valleys) correspond to replication initiation and termination zones. Regions between those peaks and valleys are characterised by steep slopes, whose sign (rep (+) or rep(−)) indicates whether the leading strand is replicated into the right (right-replicating) or left direction (left-replicating) when the DNA is viewed in standard orientation, respectively. To partition the genome into non-overlapping right and left replicating regions, we computed the mean of slopes from Repli-seq profiles of five cell lines (GM12818, K564, Hela, Huvec and Hepg2) using finite differences. We marked regions with a plus (+) if the slope was positive (and therefore right-replicating) and with minus (−) if the slope was negative (and henceforth left-replicating). To confidently assign these states, we required that the absolute value of the mean of slopes was at least larger than two times its standard deviations, otherwise we assigned the unknown (*) state to the respective region. Using this convention, a C > A variant in a rep(+) region corresponds to a template C for leading strand DNA synthesis (and a template G for lagging strand). Subsequent assignment of single base substitutions to leading and lagging strand is analogous to the procedure we used for transcription strand assignment, and adds another dimension of size of three to the count tensor ($\mathbf{C}^{SNV} \in \mathbb{N}_0^{3 \times 3 \times p \times n}$).

*Nucleosomal states*. To assign single base substitutions to minor grooves facing away from and towards histones, and linker regions between nucleosomes, we used nucleosome dyad (midpoint) positions of human lymphoblastoid cell lines mapped in MNase cut efficiency experiments[23]. Although nucleosomal DNA binding is mediated by non-sequence specific minor groove-histone interactions, histone bound DNA features 5 bp AT-rich (minor in) followed by 5 bp GC-rich (minor out) DNA, as this composition bends the molecule favourably, while its characteristic structure may lead to differential susceptibility for mutational processes. Therefore, we partitioned nucleosomal DNA by first adding 7 bp to both sides of a dyad, and assigning the following 10 alternating 5 bp DNA stretches to minor out and minor in DNA, followed by a linker region with a maximum of 58 bp. Subsequent assignment of SNVs to these states adds another dimension of size four to the count tensor ($\mathbf{C}^{SNV} \in \mathbb{N}_0^{3 \times 3 \times 4 \times p \times n}$).

*Epigenetic states*. To assign single base substitutions to different epigenetic environments, we used functional annotations from the 15-state ChromHMM model provided by the Roadmap epigenomics consortium[32], which integrates multiple chromatin datasets such as ChIP-seq data of various histone modifications. To find state annotations that are robust across all cancer tissues, we defined an epigenetic consensus state by combining state annotations from 127 different Roadmap cell lines. Here, we required that at least 70% of the cell lines agreed in the Chrom-HMM state to accept the state for a given genomic region. Partitioning SNVs by Chrom-HMM states adds another dimension of size 16 to the count tensor ($\mathbf{C}^{SNV} \in \mathbb{N}_0^{3 \times 3 \times 16 \times 4 \times p \times n}$).

Currently available data does not allow to obtain a comprehensive set of epigenetic annotations matched to the cell of origin for every cancer type. Using a consensus annotation assigns many regions to a variable (NA) state. While it is currently possible to match 31/37 PCAWG cancer types to cell lines closely corresponding to the presumed cell of origin, we note that only 9,870,018/48,329,388 mutations change category using this partially matched annotation (Supplementary Fig. 7).

*Clustered mutations*. To identify clustered single base substitutions, we used inter mutation distances ($Y_k$ in bp) between consecutive mutations on a chromosome as observations for a two state ($X_k =$ clustered, unclustered) hidden markov model with initial/transition distribution

$$p_{X_1}(x_1) = \begin{cases} 0.01 & \text{if } x_1 \text{ clustered} \\ 0.99 & \text{if } x_1 \text{ unclustered} \end{cases} \quad p_{X_{k+1}|X_k}(x_{k+1}|x_k) \begin{cases} 0.99 & \text{if } x_{k+1} = x_k \\ 0.01 & \text{if } x_{k+1} \neq x_k \end{cases} \quad (1)$$

and observation distribution

$$p_{Y_k|X_k}(y_k|x_k) = \begin{cases} \text{Geom}(p = 1/100) & \text{if } x_k \text{ clustered} \\ \text{Geom}(p = (\frac{1}{n}\sum_{k=1}^n y_k)^{-1}) & \text{if } x_k \text{ unclustered.} \end{cases} \quad (2)$$

We then computed the maximum a posteriori (MAP) state using the Viterbi algorithm to assign to each mutation the state clustered or unclustered, respectively. All steps described in this section were performed in R 3.4 (Packages: BiocInstaller/Bioconductor (1.24.0), Biostrings (2.42.1), BSgenome (1.42.0), GenomicRanges (1.26.4), VariantAnnotation (1.20.3), rhdf5 (2.18.0)).

**Signature tensor**. In mutational signature analysis, NMF is used to decompose a catalogue of cancer genomes **C** to a set of mutational signatures **S** and their

constituent activities or exposures $\mathbf{E}$. This operation can be compactly expressed as

$$\mathbb{E}[\mathbf{C}] = \mathbf{S} \times \mathbf{E} \text{ where } \mathbf{C} \in \mathbb{N}_0^{p \times n}, \mathbf{S} \in \mathbb{R}_+^{p \times s}, \text{ and } \mathbf{E} \in \mathbb{R}_+^{s \times n} \qquad (3)$$

where $p$ is the number of mutation types (usually $p = 96$), $n$ the number of cancer genomes and $s$ the number of mutational signatures.

Similarly, TensorSignatures identifies a low dimensional representation of a mutation count tensor, but decomposes it to mutational spectra for coding and template strand, leading and lagging strand, and signature specific multiplicative factors quantifying the propensities of mutational processes within specific genomic contexts. This is conceptually similar to a Tucker decomposition, a multidimensional generalisation of a single value decomposition. Yet it has the advantage over the latter that the extracted factors each have a defined biological meaning.

To enable strand specific extraction of mutational spectra requires to increase the dimensionality of the $p \times s$ sized signature matrix. To understand this, consider that two $p \times s$ matrices are at least needed to represent spectra for coding (C) and template (T) strand, suggesting a three dimensional ($2 \times p \times s$) signature representation. Our model, however, also considers replication, which adds another dimension of size two for leading (L) and lagging (G) strand, and thus we represent mutational spectra in the four dimensional core signature tensor $\mathbf{T_0} \in \mathbb{R}^{2 \times 2 \times p \times s}$

$$\mathbf{T_0} = \begin{bmatrix} \mathbf{T_0^{C/L}} & \mathbf{T_0^{C/G}} \\ \mathbf{T_0^{T/L}} & \mathbf{T_0^{T/G}} \end{bmatrix} \text{ where } \mathbf{T_0^{C/L}}, \mathbf{T_0^{C/G}}, \mathbf{T_0^{T/L}}, \mathbf{T_0^{T/G}} \in \mathbb{R}_+^{p \times s}. \qquad (4)$$

The mutation spectra $\mathbf{T_0^{·/·}}$ are normalised to 1 for each signature s, i.e., $\sum_{i=1}^{p} (\mathbf{T_0^{·/·}})_{is} = 1 \, \forall s$. However, the mutation count tensor also contains mutations from genomic regions for which strand assignment was not applicable. To still use these data for the factorisation, we map such counts to a linear combination of $\mathbf{T_0}$'s sub matrices. This is enabled by stacking strand specific $p \times s$ matrices of the core signature tensor. For example, coding strand mutations for which replicational strand assignment was not applicable, are mapped to a linear combination of both coding strand specific sub matrices $\mathbf{T_0^{C/L}}$ and $\mathbf{T_0^{C/G}}$. Stacking sub matrices of $\mathbf{T_0}$ results in $\mathbf{T_1} \in \mathbb{R}_+^{3 \times 3 \times p \times s}$

$$\mathbf{T_1} = \begin{bmatrix} \mathbf{T_0^{C/L}} & \mathbf{T_0^{C/G}} & \frac{1}{2}(\mathbf{T_0^{C/L}} + \mathbf{T_0^{C/G}}) \\ \mathbf{T_0^{T/L}} & \mathbf{T_0^{T/G}} & \frac{1}{2}(\mathbf{T_0^{T/L}} + \mathbf{T_0^{T/G}}) \\ \frac{1}{2}(\mathbf{T_0^{C/L}} + \mathbf{T_0^{T/L}}) & \frac{1}{2}(\mathbf{T_0^{C/G}} + \mathbf{T_0^{T/G}}) & \mathbf{T_0^{avg.}} \end{bmatrix} \qquad (5)$$

where $\mathbf{T_0^{avg}} = \frac{1}{4}(\mathbf{T_0^{C/L}} + \mathbf{T_0^{T/L}} + \mathbf{T_0^{C/G}} + \mathbf{T_0^{T/G}})$.

*Tensor factors.* We use the term *tensor factor* for variables of the model that are factored into the signature tensor to quantify different genomic properties of a mutational signature. The key idea is to express a mutational process in terms of a product of strand specific spectra and a set of scalars, which modulate the magnitude of spectra dependent on the genomic state combination presented in the count tensor. However, to understand how tensor factors enter the factorisation, it is necessary to introduce the concept of broadcasting, which is the process of making tensors with different shapes compatible for arithmetic operations.

It is important to realise that it is possible to increase the number of dimensions of a tensor by prepending their shapes with ones. For example, a three dimensional tensor $\mathbf{X}$ of shape $\mathbb{R}_+^{2 \times 3 \times 5}$ has 2 rows, 3 columns and a depth of 5. However, we could reshape $\mathbf{X}$ to $\mathbb{R}_+^{1 \times 3 \times 1 \times 2 \times 5}$, or $\mathbb{R}_+^{2 \times 3 \times 1 \times 5 \times 1}$, which would eventually change the order of values in the array, but not its content. These extra (empty) dimensions of $\mathbf{X}$ are called singletons or degenerates, and are required to make entities of different dimensionality compatible for arithmetic operations via broadcasting. To understand this, consider the following example

$$\begin{bmatrix} 1 & 2 \end{bmatrix}_{\mathbb{R}^{1 \times 2}} \odot \begin{bmatrix} 3 \\ 4 \end{bmatrix}_{\mathbb{R}^{2 \times 1}} = \underbrace{\begin{bmatrix} 1 & 2 \\ 1 & 2 \end{bmatrix} \cdot \begin{bmatrix} 3 & 3 \\ 4 & 4 \end{bmatrix}}_{\text{Broadcasting and element} - \text{wise multiplication.}} = \begin{bmatrix} 3 & 6 \\ 4 & 8 \end{bmatrix}_{\mathbb{R}^{2 \times 2}}. \qquad (6)$$

The $\odot$ operator first copies the elements along their singleton axes such that the shape of both resulting arrays match, and then performs element-wise multiplication as indicated by the $\cdot$ symbol. This concept is similar to the tensor product $\otimes$ for vectors, but also applies to higher dimensional arrays, although this requires to define the shapes of all tensors carefully. For example if $\mathbf{F} \in \mathbb{R}^{2 \times 2}$ and $\mathbf{H} \in \mathbb{R}^{1 \times 1 \times 3}$ then $\mathbf{F} \odot \mathbf{H}$ is an invalid operation, however, if $\mathbf{G} \in \mathbb{R}^{2 \times 2 \times 1}$, then $(\mathbf{G} \odot \mathbf{H}) \in \mathbb{R}^{2 \times 2 \times 3}$ is valid. Also, note that such operations are not necessarily commutative.

**Transcriptional and replicational strand biases.** To quantify spectral asymmetries in context of transcription and replication, we introduce two vectors $\mathbf{b_t}, \mathbf{b_r} \in \mathbb{R}_+^{1 \times s}$, stack and reshape them such that the resulting bias tensor

$\mathbf{B} \in \mathbb{R}_+^{3 \times 3 \times 1 \times s}$,

$$\mathbf{B} = \begin{bmatrix} \mathbf{b_r} \cdot \mathbf{b_t} & \mathbf{b_r^{-1}} \cdot \mathbf{b_t} & 1 \cdot \mathbf{b_t} \\ \mathbf{b_r} \cdot \mathbf{b_t^{-1}} & \mathbf{b_r^{-1}} \cdot \mathbf{b_t^{-1}} & 1 \cdot \mathbf{b_t^{-1}} \\ \mathbf{b_r} \cdot 1 & \mathbf{b_r^{-1}} \cdot 1 & 1 \cdot 1 \end{bmatrix}, \qquad (7)$$

matches the shape of $\mathbf{T_1}$. Also, note that signs of $\mathbf{b_t}$ and $\mathbf{b_r}$ are chosen such that positive values correspond to a bias towards coding and leading strand, while negative values indicate shifts towards template and lagging strand.

**Signature activities in transcribed/untranscribed and early/late replicating regions.** To assess the activity of mutational processes in transcribed versus untranscribed, and early versus late replicating regions, we introduce two additional scalars per signature represented in two vectors $\mathbf{a_t}$ and $\mathbf{a_r} \in \mathbb{R}_+^{1 \times s}$. Both vectors are stacked and reshaped to match the shape of $\mathbf{T_1}$,

$$\mathbf{A} = \begin{bmatrix} \mathbf{a_t} \cdot \mathbf{a_r} & \mathbf{a_t} \cdot \mathbf{a_r} & \mathbf{a_t} \\ \mathbf{a_t} \cdot \mathbf{a_r} & \mathbf{a_t} \cdot \mathbf{a_r} & \mathbf{a_t} \\ \mathbf{a_r} & \mathbf{a_r} & 1 \end{bmatrix}. \qquad (8)$$

**Mutational composition.** To quantify the percentage of SNVs and other mutation types requires another $1 \times s$ sized vector $\mathbf{m}$, satisfying the constraint $0 \le \mathbf{m}_i \le 1$ for $i = 1, \ldots, s$. In order to include $\mathbf{m}$ in the tensor factorisation we reshape the vector to $\mathbf{M} \in \mathbb{R}_+^{1 \times 1 \times 1 \times s}$, while $(1 - \mathbf{m})$ is multiplied with the secondary signature matrix $\mathbf{S}$.

**The strand-specific signature tensor.** We define the strand-specific signature tensor as

$$\mathbf{T_{strand}} := \mathbf{T_1} \odot \mathbf{B} \odot \mathbf{A} \odot \mathbf{M}, \text{ where } \mathbf{T_2} = \mathbb{R}_+^{3 \times 3 \times p \times s}, \qquad (9)$$

which therefore subsumes all parameters to describe a mutational process with regard to transcription and replication, and quantifies to what extent the signature is composed of SNVs. To understand this, consider the entry of the count tensor representative for coding strand mutations, e.g. $(\mathbf{T_{strand}})_{13..} = \mathbf{b_t} \odot \mathbf{a_t} \odot \mathbf{m} \odot \frac{1}{2}(T_0^{C/G} + T_0^{C/T})$, which explicitly states how the low dimensional tensor factors for transcription are broadcasted into the signature tensor.

**Signature activities for nucleosomal, epigenetic and clustering states.** The strand-specific signature tensor $\mathbf{T_{strand}}$ can be considered as the basic building block of the signature tensor, as we instantiate "copies" of $\mathbf{T_{strand}}$ by broadcasting scalar variables for each genomic state and signature along their respective dimensions. To understand this, recall that we, for example, split SNVs in $t = 3$ nucleosome states (minor in, minor out and linker regions). However, since SNVs may also fall into regions with no nucleosomal occupancy, we distributed mutations across $t + 1 = 4$ states in the corresponding dimension of the mutation count tensor. To fit parameters assessing the activity of each signature along these states, we initialise a matrix $\mathbf{k} \in \mathbb{R}^{(t+1) \times s}$, which can be considered as a composite of a $1 \times s$ constant vector ($\mathbf{k}_{1i} = 1$ for $i = 1, \ldots, s$) and a $t \times s$ matrix of state variables, allowing the model to adjust these parameters with respect to the first row, which corresponds to the non-nucleosomal mutations (baseline). To include these parameters in the factorisation we first introduce a singleton dimensions in the strand specific signature tensor such that $\mathbf{T_{strand}} \in \mathbb{R}_+^{3 \times 3 \times 1 \times p \times s}$, and reshape $\mathbf{k}$ to match the dimensionality of $\mathbf{T_{strand}}$,

$$\mathbf{k} \in \mathbb{R}_+^{(t+1) \times s} \Rightarrow \mathbf{K} \in \mathbb{R}_+^{1 \times 1 \times (t+1) \times 1 \times s}. \qquad (10)$$

Both tensors have now the right shape such that element wise multiplication with broadcasting is valid

$$\mathbf{T} = \mathbf{T_{strand}} \odot \mathbf{K} \text{ where } \mathbf{T} \in \mathbb{R}_+^{3 \times 3 \times (t+1) \times p \times s}. \qquad (11)$$

We proceed similarly for all remaining genomic properties such as activities along epigenetic domains, and clustering propensities. Generally, to assess $l$ genomic properties, we first introduce $l$ singleton dimensions to the strand-specific signature tensor $\mathbf{T_{strand}}$, instantiate $l$ matrices $\mathbf{k_j} \in \mathbb{R}_+^{(t_j+1) \times s}$ for $j = 1, \ldots, l$ each with $t_j$ states, reshape them appropriately to tensor factors $\mathbf{K_j}$, and broadcast them into the strand specific signature tensor $\mathbf{T_2}$. Here, we introduced new dimensions for epigenetic domains (epi), nucleosomal location (nuc) and clustering propensities (clu), and thus we reshaped the strand specific signature tensor to $\mathbf{T_{strand}} \in \mathbb{R}_+^{3 \times 3 \times 1 \times 1 \times 1 \times p \times s}$, instantiated $\mathbf{k_{epi}} \in \mathbb{R}_+^{16 \times s}$, $\mathbf{k_{nuc}} \in \mathbb{R}_+^{4 \times s}$ and $\mathbf{k_{clu}} \in \mathbb{R}_+^{2 \times s}$ and computed

$$\mathbf{T} = \mathbf{T_{strand}} \odot \mathbf{K_{epi}} \odot \mathbf{K_{nuc}} \odot \mathbf{K_{clu}} \text{ where } \mathbf{T} \in \mathbb{R}_+^{3 \times 3 \times 16 \times 4 \times 2 \times p \times s} \qquad (12)$$

to obtain the final signature tensor $\mathbf{T}$.

**Model assumptions.** The model assumes that the expected values of $\mathbf{C^{SNV}}$ and $\mathbf{C^{other}}$ are determined by the inner product of the signature tensor $\mathbf{T}$ (using the convention that $\times$ is taken over the last dimension of the array on its left - denoting each different signature - and the first dimension of the array on its right) and the

exposure matrix $\mathbf{E}$ and similarly for the non-SNV signature matrix $\mathbf{S}$ and the same exposure matrix $\mathbf{E}$

$$\mathbb{E}[\mathbf{C}^{\text{SNV}}] = \mathbf{T} \times \mathbf{E} \quad \text{and} \quad \mathbb{E}[\mathbf{C}^{\text{other}}] = \underbrace{(\mathbf{S_0} \odot (1-\mathbf{m}))}_{\mathbf{S}} \times \mathbf{E}. \tag{13}$$

To prevent over segmentation and ensure a robust fit of signatures, we assume that the data follows a negative binomial distribution with mean $\mathbf{T} \times \mathbf{E}$ and $\mathbf{S} \times \mathbf{E}$, and dispersion $\tau$

$$\mathbf{C}^{\text{SNV}}_{i\ldots n} \sim \text{NB}((\mathbf{T} \times \mathbf{E})_{i\ldots n}, \tau) \quad \text{and} \quad \mathbf{C}^{\text{other}}_{mn} \sim \text{NB}((\mathbf{S} \times \mathbf{E})_{mn}, \tau). \tag{14}$$

We use the Tensorflow framework to find the maximum likelihood estimates (MLE) $\hat{\mathbf{T}}, \hat{\mathbf{S}}, \hat{\mathbf{E}}$ for $\mathbf{T}, \mathbf{S}$ and $\mathbf{E}$ respectively using the parametrisation defined in the previous section. We initialise the parameters of the model with values drawn from a truncated normal distribution and compute $\hat{\mathbf{T}} \times \hat{\mathbf{E}}$ and $\hat{\mathbf{S}} \times \hat{\mathbf{E}}$ which are fed into the negative binomial likelihood function

$$\mathcal{L}^{\text{SNV}}(\mathbf{C}^{\text{SNV}}_{i\ldots n}; (\mathbf{T} \times \mathbf{E})_{i\ldots n}, \tau) = \prod_{i\ldots n} \frac{\Gamma(\tau + \mathbf{C}^{\text{SNV}}_{i\ldots n})}{\Gamma(\tau)\mathbf{C}^{\text{SNV}}_{i\ldots n}!} \left(\frac{\tau}{\tau + \mathbf{C}^{\text{SNV}}_{i\ldots n}}\right)^{\tau} \left(\frac{(\mathbf{T} \times \mathbf{E})_{i\ldots n}}{\tau + (\mathbf{T} \times \mathbf{E})_{i\ldots n}}\right)^{\mathbf{C}^{\text{SNV}}_{i\ldots n}} \tag{15}$$

and

$$\mathcal{L}^{\text{other}}(\mathbf{C}^{\text{other}}_{mn}; (\mathbf{S} \times \mathbf{E})_{mn}, \tau) = \prod_{mn} \frac{\Gamma(\tau + \mathbf{C}^{\text{other}}_{mn})}{\Gamma(\tau)\mathbf{C}^{\text{other}}_{mn}!} \left(\frac{\tau}{\tau + \mathbf{C}^{\text{other}}_{mn}}\right)^{\tau} \left(\frac{(\mathbf{S} \times \mathbf{E})_{mn}}{\tau + (\mathbf{S} \times \mathbf{E})_{mn}}\right)^{\mathbf{C}^{\text{other}}_{mn}}. \tag{16}$$

The total log likelihood $\log \mathcal{L}$ is then given by the sum of individual log likelihoods

$$\log \mathcal{L}(\mathbf{C}^{\text{SNV}}, \mathbf{C}^{\text{other}}; \mathbf{T}, \mathbf{S}, \mathbf{E}, \tau) = \log \mathcal{L}^{\text{SNV}}(\mathbf{C}^{\text{SNV}}_{i\ldots n}; (\mathbf{T} \times \mathbf{E})_{i\ldots n}, \tau) + \log \mathcal{L}^{\text{other}}(\mathbf{C}^{\text{other}}_{mn}; (\mathbf{S} \times \mathbf{E})_{mn}, \tau) \tag{17}$$

and thus the optimisation problem boils down to maximise the total log likelihood (or equivalently to minimise the negative total log likelihood)

$$\hat{\mathbf{T}}, \hat{\mathbf{S}}, \hat{\mathbf{E}} = \text{argmin}_{\mathbf{T},\mathbf{S},\mathbf{E}}\{-\log \mathcal{L}(\mathbf{C}^{\text{SNV}}, \mathbf{C}^{\text{other}}; \mathbf{T}, \mathbf{S}, \mathbf{E}, \tau)\}. \tag{18}$$

Moreover, inferring $\hat{\mathbf{T}}, \hat{\mathbf{S}}$, and $\hat{\mathbf{E}}$ enables us to calculate log likelihood of the MLE

$$\log \hat{\mathcal{L}} = \log \mathcal{L}(\mathbf{C}^{\text{SNV}}, \mathbf{C}^{\text{other}}; \hat{\mathbf{T}}, \hat{\mathbf{S}}, \hat{\mathbf{E}}, \tau). \tag{19}$$

To calculate the value of each parameter in the model, we minimise the negative total log likelihood using an ADAM Grad optimiser with an exponentially decreasing learning rate of 0.1 and approximately 50,000 epochs. We implemented TensorSignature in Python 3.6 (Packages: h5py (2.7.1), ipython (6.3.1), matplotlib (3.0.2), numpy (1.16.1), pandas (0.22.0), scikit-learn (0.19.1), scipy (1.0.1), seaborn (0.9.0), scipy (1.0.1), tensorflow (1.4.1), tensorsignatures (0.4.0), tqdm (4.28.1)).

**Model selection**. To select the appropriate number of signatures for a model with dispersion $\tau$ and dataset, we compute for each rank $s$ the Bayesian Information Criterion (BIC)

$$\text{BIC}_{\tau}(s) = \log(n) \cdot k(s) - 2 \cdot \log \hat{\mathcal{L}}, \tag{20}$$

where $n$ is the number of observations (total number of counts in $\mathbf{C}^{\text{SNV}}$ and $\mathbf{C}^{\text{other}}$), $k(s)$ represents number of parameters in the model (which depends on the rank $s$), and $\log \hat{\mathcal{L}}$ is the log-likelihood of the MLE. The BIC tries to find a trade-off between the log-likelihood and the number of parameters in the model; chosen is the rank which minimises the BIC.

**Bootstrap confidence intervals**. To compute bootstrap confidence intervals (CIs) for inferred parameters, we randomly select $\frac{2}{3}$ of the samples in the dataset, initialise the model with the MLE for $\hat{\mathbf{T}}$ and $\hat{\mathbf{S}}$ while randomly perturbing the 10% of their estimates, and subsequently refit $\hat{\mathbf{T}}, \hat{\mathbf{S}}$ and $\hat{\mathbf{E}}$ to the subset of samples. Initialising the parameters with the MLE results from computational constraints, as this step needs to be repeated for 300–500 times to obtain representative distributions of the parameter space. Next, we match refitted signatures to the MLE reference by computing pairwise cosine distances, and accept bootstrap samples if the total variation distance between the bootstrap candidate and the reference is smaller than 0.2. Finally, we compute 5% and 95% percentiles on accepted bootstrap samples to indicate the CIs of our inference.

**XPC genomes**. Somatic single nucleotide variants were called from .bam files were called as described in[48]. Subsequently these were aggregated into a mutation count tensor as described above.

**Comparing TensorSignatures to conventional NMF**
*Extracting signature properties using conventional NMF*. We tried to quantify the genomic properties of mutational signatures using a less principled approach by simulating mutational signatures plus their genomic properties, sample exposures, and resulting mutation counts. To recover mutational signatures and corresponding sample exposures, we factorised (summed) mutation counts of simulated data using 96-trinucleotide channels only. To determine the strand biases and signatures activities across genomic states, we fixed the spectra of previously identified signatures, and refitted their exposures to the count matrix containing the mutations of a specific state only (eg. template strand mutations, TssA). To obtain a scalar parameter descriptive signature properties, we regressed state specific exposures to their respective baseline exposures (eg. exposures of template strand mutations against exposures of unassigned mutations) and compared obtained regression coefficients with the equivalent parameter of the tensor factorisation and ground truth. To assess the error, we computed the vector 2-norm and cosine similarity for strand biases, genomic activities and exposures, and signature spectra respectively. We performed this experiment for datasets with sizes (100, 1000, 10000) and different numbers of mutations per sample (100, 1000, 10000). Note, in this approach it is not possible to recover signature activities in untranscribed/transcribed and early/late replicating regions (indicated as "Amplitudes" in the following plot).

Our simulations revealed increasing relative errors for all assessed parameters as sample size and mutation loads increase (Supplementary Fig. 2, middle panel). To understand this, consider that only TensorSignatures may leverage the additional information encoded in the tensor representation of larger datasets to improve the estimates of signature defining properties such as strand biases and genomic activities. Although it is possible to find reasonable parameter estimates by fitting signature spectra first and subsequently regressing out the effect of genomic determinants, absolute errors are always larger at similar samples sizes and mutation loads (Supplementary Fig. 2, lower panel).

*Assigning single base substitutions to their source signature with maximum a posteriori approaches*. To assess TensorSignatures' ability of assigning mutations to their appropriate source signature, we designed a simulation experiment in which we used two very similar signatures (TS05 and TS06) to simulate a mutation count tensor. We then applied conventional NMF on the marginalised (summed) count tensor and determined the maximum a posteriori (MAP) signature for each trinucleotide context in each simulated sample as described in[19].

Absolute errors (vector 2-norm) of post-hoc assigned mutations increase as the number of mutations per sample get larger, while the predictions of the equivalent tensor factorisation become more accurate. This is to be expected as the post-hoc signature posterior probability is only conditioned on the mutation type and the sample exposure. Furthermore, shown results are likely to underestimate errors as our simulations/inferences were performed using only two signatures, and thus correct signature assignment is likely to happen by chance (Supplementary Fig. 3).

*Stability of solutions*. Another challenge in mutational signature analysis is the problem of unambiguously associating other variant types to their respective mutational processes. Common practice is to perform independent NMFs on each variant type, and to subsequently match subtype specific signatures to their SNV correlate by assessing exposures. In contrast, TensorSignatures decomposes SNV and other mutation type counts simultaneously, thus circumventing the problem of post-hoc associating different mutation types, and delivering a more robust signature inference by pooling evidence from the entire mutational imprint.

To illustrate this, we ran independent NMFs on SNV and other mutation count matrices of the PCAWG dataset. To match resulting mutational spectra, we computed the correlation coefficients of their exposures and paired highest correlating signatures. We repeated these steps 50 times to obtain a set of 50 initialisations of paired mutational signatures (SNV and other mutation types), and compared the stability of these solutions with TensorSignatures decompositions by computing the silhouette scores across several ranks (Supplementary Fig. 4).

Our results indicate a higher stability of TensorSignatures solutions across all tested ranks implying that the tensor framework more consistently reproduced SNV and their accompanying other mutation type spectra.

**Reporting summary**. Further information on research design is available in the Nature Research Reporting Summary linked to this article.

## Data availability

PCAWG WGS primary tumour data is available under restricted access, access can be obtained at [http://dcc.icgc.org/pcawg]. Most data of ICGC and TCGA projects are in an open tier which does not require access approval. For full access, researchers will need to apply to the TCGA Data Access Committee (DAC) via dbGaP [https://dbgap.ncbi.nlm.nih.gov/aa/wga.cgi?page=login] for access to the TCGA portion of the dataset, and to the ICGC Data Access Compliance Office (DACO; [http://icgc.org/daco]) for the ICGC portion. In addition, to access somatic single nucleotide variants derived from TCGA donors, researchers will also need to obtain dbGaP authorisation. The HMF WGS metastatic tumour data data is available under restricted access, access can be obtained at [https://www.hartwigmedicalfoundation.nl/applying-for-data/]. A data request can be initiated by explaining the intended use of the requested data. Upon data access board approval, a standard license agreement without restrictions regarding intellectual property resulting from the data analysis needs to be signed by an official organisation representative before access to the data are granted. The *XPC* WGS genome data is available under restricted access, access can be obtained under the accession number phs000830.v1.p1 [https://www.ncbi.nlm.nih.gov/projects/gap/cgi-bin/study.cgi?study_id=phs000830.v1.p1]. Data access

may only be granted to researchers at a level equivalent to a tenure-track professor or senior scientist. For further guidance on data access, please see [https://www.ncbi.nlm.nih.gov/projects/gap/cgi-bin/GetPdf.cgi?document_name=GeneralAAInstructions.pdf]. The Gencode v19 transcription annotation data used in this study are available at [https://www.gencodegenes.org/human/release_19.html]. The Repli-seq data for GM12818, K564, Hela, Huvec and Hepg2 cell lines used in this study are available at [https://www.encodeproject.org/search/?type=Experiment&assay_title=Repli-seq]. The 15-state ChromHMM annotation data used in this study are available at [https://egg2.wustl.edu/roadmap/web_portal/chr_state_learning.html]. The nucleosome MNase dataset used in this study are available at [https://egg2.wustl.edu/roadmap/web_portal/chr_state_learning.html]. The remaining data are available within the Article, Supplementary Information or available from the authors upon request. Source data are provided with this paper.

## Code availability

TensorSignatures' source code is available at [http://github.com/gerstung-lab/tensorsignatures] and as a pypi package "tensorsignatures"[51]. This repository contains code for data preprocessing, genomic annotation and signature discovery and fitting. TensorSignatures can also be run as TensorSignaturesOnline, a web application accessible under [http://gerstung-lab.github.io/tensorsignatures], that enables users to analyse their VCF data by attributing variants to a set of predefined TensorSignatures. The usage of the web application requires an online registration, which enables the access to an upload form to which VCF data may be uploaded and subsequently analysed. All analyses were performed using Python 3.6 (Packages: h5py (2.7.1), ipython (6.3.1), matplotlib (3.0.2), numpy (1.16.1), pandas (0.22.0), scikit-learn (0.19.1), scipy (1.0.1), seaborn (0.9.0), scipy (1.0.1), tensorflow (1.4.1), tensorsignatures (0.4.0), tqdm (4.28.1)), or R 3.4 (Packages: BiocInstaller/Bioconductor (1.24.0), Biostrings (2.42.1), BSgenome (1.42.0), GenomicRanges (1.26.4), VariantAnnotation (1.20.3), rhdf5 (2.18.0)).

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

## Acknowledgements

We thank Oriol Pich, Santiago Gonzalez and Nuria Lopez for help in providing the genome coordinates for nucleosomes positions and variant calls for *XPC* genomes. Also, we thank Nadezda Volkova and Jose Guillherme de Almeida for commenting on our manuscript. This publication and the underlying study have been made possible partly on the basis of the data that Hartwig Medical Foundation and the Center of Personalised Cancer Treatment (CPCT) have made available to the study.

## Author contributions

H.V. conducted all bioinformatic analyses and produced the figures. A.v.H. and E.C. curated HMF data and provided computing resources for HMF data analysis by H.V. M.G. conceived and supervised the analysis and developed code for categorising mutations. H.V. and M.G. wrote the manuscript with input from A.v.H. and E.C.

## Funding

## Competing interests

The authors declare no competing interests.
