## [Peer Review File · Nature Communications]

REVIEWERS' COMMENTS

Reviewer #1 (Remarks to the Author):

-

Reviewer #2 (Remarks to the Author):

With the exception of my comments about TensorSignatures implementation, the authors have responded to my original concerns by mostly ignoring them. After re-reviewing the manuscript and the authors' responses, I believe that this paper should be rejected. As noted by the other reviewers and confirmed by the authors, this manuscript is mostly a methodological advancement which does not aim to provide novel biological results. Unfortunately, TensorSignatures does not seem technically sound as it misses a number of well described signatures, which are found in the PCAWG dataset, with strong properties which TensorSignatures aims to describe. For example, TensorSignatures misses the signature of aflatoxin which has strong transcriptional strand bias and is found in the PCAWG liver cancer. This is not only a limitation but also a major flaw in TensorSignatures that may result in misreporting properties in future analysis. I have listed below my technical concerns. This is the same list as in the previous round of review as most of these concerns, with the exception of the tool's implementation, were not addressed.

1) Missing known signatures

The authors suggest addressing the missing signatures by expanding their discussion and listing current limitations of TensorSignatures. I do not think this is acceptable as missing these signatures shows a possible underlying problem which may be biasing subsequent analysis. Further, TensorSignatures is not only missing signatures with unknown etiology but also the signatures of aflatoxin, platinum therapy, and failure of base excision repair due to NTHL1 mutations (with at least 3 other signatures with unknown etiology also missing). The signatures of aflatoxin, platinum therapy, and NTHL1 failure are described in multiple cohorts, experimentally validated in model systems, and found in the ~2700 PCAWG samples examined by the authors.

The issue is not only in that these known signatures are not detected. Some of these signatures have well understood properties, for example, strong transcriptional strand bias for aflatoxin and platinum therapy. TensorSignatures misassigns the mutations of these signature to other tensor factors which likely results in misreporting of transcriptional strand bias.

The authors claim to have performed "a fair comparison between TensorSignatures and SigProfiler on 2778 PCAWG genomes". This is simply not the case as, for example, the authors did not use the official set of released signatures but re-ran SigProfiler with uncertain parameters. Importantly, the

PCAWG mutational signatures manuscript generated a number of synthetic datasets with known ground-truth signatures. If the authors want fair comparison, they should run TensorSignatures on these data (<https://www.synapse.org/#!Synapse:syn18497223>) and compare them to the results of both SigProfiler and SignatureAnalyzer.

2) Incorporating additional features into a tensor

The authors have not addressed this concern. Their response is: “While we agree that it would be preferable to have an epigenetic annotation matched to every cancer type, or ideally every cancer, this is currently infeasible for all cancers and all annotations.” Indeed, they have added a supplementary figure 7a where cancer types are matched to cell lines. However, this matching is simply not reasonable. The authors can do much better by using the new release of ENCODE for epigenomic annotation and replication timing. NucPosDB can be used for MNase data. Currently, without at least attempting to correctly match tissues with cancer types, I do not find the results biologically meaningful.

3) Nonnegative factorization of the constructed tensor

In no way, the authors’ response provides convincing evidence that TensorSignatures adds additional value as a method or “produces more accurate results” (particularly concerning statement considering the number of signatures that are not detected by TensorSignatures). As stated by the authors, performing a Tucker tensor decomposition will indeed significantly reduce the number of parameters that need to be fitted. This reduction is possible due to making assumptions of the structure of the data represented by the different tensor dimensions. Whether or not such assumptions are warranted is a philosophical question as “all models are wrong, but some are useful”. The question the authors need to answer is whether their method is useful when compared to previous approaches. Specifically, features of mutational signatures have been reported by a number of previous studies. The authors should compare their findings to at least one such analysis. For example, somewhat similar analysis was previously reported in Nature Communications for 560 breast cancers using traditional nonnegative matrix factorization (PMID: 27136393). What is the value that TensorSignatures brings beyond such prior approaches?

4) Implementation of the tool

The implementation of the tool and its documentation are much improved. I have been able to install TensorSignatures and to run it.

Reviewer #3 (Remarks to the Author):

The authors have addressed my comments and have improved the manuscript in the new revision. As a minor comment around presentation, I thought that the dichotomy of discovery (PCAWG) and validation (HMF) of various signatures could be explained more clearly throughout the results section. Specific statements in the text could be added to support findings of signatures and help future readers. I also felt that the manuscript was dense and long but I will leave that up to the discussion in the other where specific comments around novelty are discussed.

Reviewer #4 (Remarks to the Author):

Draft comments on NCOMMS-20-30726A - Learning mutational signatures and their multidimensional genomic properties with TensorSignatures
Rozen, 6 Dec 2020

I was asked to provide an opinion on whether responses to several concerns raised by Reviewer 2 were adequate. I did not do a full review.

In summary, with one exception, I believe the responses to the Reviewer's concerns were adequate.

In a larger context, I would point out that, for discovery (extraction) of mutational signatures, the aphorism "All models are wrong, but some are useful" very much applies, as was also noted by Reviewer 2. TensorSignatures introduces a new class of models that represent a substantial conceptual advance, and the analyses reported in the manuscript indicate that these models are likely to be useful.

All methods for discovery of mutational signatures generate imperfect models and require *substantial* human interpretation in the light of all available evidence. None of the methods for discovering mutational signatures can automatically and algorithmically provide a unique "true" model. They can only provide one of many possible models that need to be further assessed.

In view of this, I am not concerned about signatures in Alexandrov et al. 2020 (<https://www.nature.com/articles/s41586-020-1943-3>) that were not detected by TensorSignatures. This is for two reasons. (1) In Alexandrov et al. and in COSMIC, some signatures were indeed pulled in from tumors that were not in the core PCAWG "platinum" data set of ~2780 genomes that the Vöhringer et al. manuscript analyzes. (2) Unsupervised learning approaches (including all methods for mutational signature discovery) require a tradeoff between sensitivity and specificity. Indeed, assessing this tradeoff is part of the human judgement needed for interpreting the results of these methods. I believe that the TensorSignatures analysis struck a reasonable balance between sensitivity and specificity.

Regarding testing against the synthetic data in Alexandrov et al., I concur with the authors that these synthetic data are not suitable: they do not have the genomic location information that TensorSignatures uses.

Regarding the Reviewer's concern that the epigenomic annotations were not as comprehensive or as up-to-date as possible, I believe that the annotations that were used are sufficient. There will always be new and better annotations. The information that can be added by these should be explored by the authors or others in the future.

Regarding the Reviewer's concern about comparison to `_results_` from previous approaches to understanding the relationships between signatures and genomic features, I do think a qualitative

discussion of the similarities and differences to what was found in e.g. Morganella et al., Nat. Commun. 7, 11383 and/or other genomic topography/landscape papers cited in the intro would add to the manuscript. It is difficult for the reader to make comparisons between TensorSignature's findings and those the previous literature for a variety of reasons. Granted that there are some challenges, including splits / merges of SBS signatures between the current manuscript and signatures used in previous studies. Nevertheless, this discussion would add to the manuscript. I do not think TensorSignatures has to be superior in every respect to previous approaches to be useful, but I do think something in the Discussion that summarizes TensorSignature's findings in the context of findings by other results would be useful to the reader.

Minor comments: Please specify the version of SigProfiler used and any non-default parameters.

Typos: references still in the old Cell style on lines 944 and 1351.

Signed: Steven G Rozen

Note to the Editor: Replies to Reviewer #2's comments have not been amended since the last submission as the comments have not changed.

REVIEWERS' COMMENTS

Reviewer #1 (Remarks to the Author):

-

Reviewer #2 (Remarks to the Author):

With the exception of my comments about TensorSignatures implementation, the authors have responded to my original concerns by mostly ignoring them. After re-reviewing the manuscript and the authors' responses, I believe that this paper should be rejected. As noted by the other reviewers and confirmed by the authors, this manuscript is mostly a methodological advancement which does not aim to provide novel biological results. Unfortunately, TensorSignatures does not seem technically sound as it misses a number of well described signatures, which are found in the PCAWG dataset, with strong properties which TensorSignatures aims to describe. For example, TensorSignatures misses the signature of aflatoxin which has strong transcriptional strand bias and is found in the PCAWG liver cancer. This is not only a limitation but also a major flaw in TensorSignatures that may result in misreporting properties in future analysis. I have listed below my technical concerns. This is the same list as in the previous round of review as most of these concerns, with the exception of the tool's implantation, were not addressed.

1) Missing known signatures

The authors suggest addressing the missing signatures by expanding their discussion and listing current limitations of TensorSignatures. I do not think this is acceptable as missing these signatures shows a possible underlying problem which may be biasing subsequent analysis. Further, TensorSignatures is not only missing signatures with unknown etiology but also the signatures of aflatoxin, platinum therapy, and failure of base excision repair due to NTHL1 mutations (with at least 3 other signatures with unknown etiology also missing). The signatures of aflatoxin, platinum therapy, and NTHL1 failure are described in multiple cohorts, experimentally validated in model systems, and found in the ~2700 PCAWG samples examined by the authors.

The issue is not only in that these known signatures are not detected. Some of these signatures have well understood properties, for example, strong transcriptional strand bias for aflatoxin and platinum therapy. TensorSignatures misassigns the mutations of these signature to other tensor factors which likely results in misreporting of transcriptional strand bias.

Aflatoxin (n=2), platinum therapy (n=13) and potential NTHL1 deficiency (n=46) occur only in a few samples of the PCAWG dataset and provide an estimated total of 0.2% of single base substitutions, making it difficult to recover them for any mutational signature algorithm to detect these processes. We note that the signature of platinum therapy is found in the HMF cohort and at higher ranks in the PCAWG cohort.

Given that the missed signatures are exclusively rare and contribute only a very small fraction of mutations, it is very unlikely that not accounting for these signatures leads to a considerable bias of the remaining signatures. (This is confirmed by simulations, which also at times miss a simulated signature but nevertheless produce unbiased estimates.)

Moreover, one should also not forget about the current status quo of estimating genomic activity by assigning individual mutations to signature, which leads to considerable confusion of effects of the mutation spectra are similar. This is exemplified by the case of the two UV signatures, which we discuss in detail in the manuscript and which are reported to both have a transcriptional strand bias in the COSMIC catalogue.

The authors claim to have performed “a fair comparison between TensorSignatures and SigProfiler on 2778 PCAWG genomes”. This is simply not the case as, for example, the authors did not use the official set of released signatures but re-ran SigProfiler with uncertain parameters. Importantly, the PCAWG mutational signatures manuscript generated a number of synthetic datasets with known ground-truth signatures. If the authors want fair comparison, they should run TensorSignatures on these data (<https://www.synapse.org/#!Synapse:syn18497223>) and compare them to the results of both SigProfiler and SignatureAnalyzer.

Throughout the manuscript we refer to the COSMIC catalogue, which is highly curated, and seen by many as the current gold standard. When it comes to the properties of the underlying algorithms, however, one has to keep in mind that the official mutational signatures reported for the PCAWG Data set were informed by an extra 1,800 cancer genomes from ICGC and other sources, thus providing extra evidence for the rare signatures. Furthermore, the underlying analysis used a bespoke set of extra rules modelling preferred signature activities in different cancer types, which are difficult to automate and generalise, rather than simply running SigProfiler across all cases. And lastly many artefacts have been manually removed. This makes it a very useful reference, but we do note that it's practically impossible to reproduce these analyses.

With TensorSignatures we did not primarily aim for discovering as many signatures as possible, but rather to develop an automated framework for studying the characteristics of recurrent signatures. As the reviewer was concerned that some known signatures were missing from our analysis in relation to the curated catalogue, we compared TensorSignatures to running SigProfiler in a similar way to demonstrate that missing rare signatures is not a limitation of the algorithm itself, but rather from not tweaking it to achieve maximal sensitivity (at the price of lower specificity resulting in split signatures and other artefacts). We summarize aforementioned points in the following section in the discussion of our manuscript.

In comparison to comparable tools for mutational signature analysis, TensorSignatures is currently not optimised to maximise the number of extracted signatures, but rather to characterise the properties of recurrent mutational processes. Compared to the curated COSMIC catalogue of mutational signatures, the automated analysis with TensorSignatures currently misses the signatures of of Aflatoxin (SBS24, n=2 samples with relative exposure > 5%), platinum therapy (SBS31, n=2; SBS35, n=11) and a signature characteristic of base excision deficiency by NTHL1 defects (SBS30, n=45), which in aggregate are estimated to

contribute 0.2% of single base substitutions in the PCAWG cohort. One reason for this discrepancy is that the COSMIC catalogue is informed by an additional 1,865 whole genome samples from other sources, which provide stronger evidence for rarely recurring mutational signatures. In order to maximise the yield of novel signatures — a somewhat different objective to characterising genomic properties of recurrent processes — a more bespoke analysis recognising each cancer type and its preferred set of mutational signatures as well as a careful assessment of potential sequencing and analysis artefacts is necessary.

The data suggested by the reviewer only provide mutation count data, but neither genomic coordinates or known ground truth about genomic activity patterns. So it is unfortunately of limited use for our purposes.

In order to clarify the discrepancy between the curated signature catalogue and our analysis we suggest to add a paragraph to the discussion (see above).

2) Incorporating additional features into a tensor

The authors have not addressed this concern. Their response is: “While we agree that it would be preferable to have an epigenetic annotation matched to every cancer type, or ideally every cancer, this is currently infeasible for all cancers and all annotations.” Indeed, they have added a supplementary figure 7a where cancer types are matched to cell lines. However, this matching is simply not reasonable. The authors can do much better by using the new release of ENCODE for epigenomic annotation and replication timing. NucPosDB can be used for MNase data. Currently, without at least attempting to correctly match tissues with cancer types, I do not find the results biologically meaningful.

We used the most recent epigenomic annotations from ENCODE available at the time when the analysis was conducted (downloaded 2019/01) and mapped them to each cancer type, wherever possible. This showed that the consensus produces a very good approximation to the partial tissue matching and if anything seems to underestimate the true effect sizes (Supplementary Figure 7). It is not clear which element of this analysis is ‘unreasonable’.

For other annotations we followed the approach of other landmark publications:

For repli-seq data we have been using a consensus across 5 different biological experiments and assigned regions far from replication sites or with variable signal as unknown orientation (*), and used these consensus regions to discover replication asymmetries in different cancer types. The same approach has been employed in (Haradhvala et al., Cell 2016).

Regarding MNase data, we used the same data used in (Pich et al., Cell 2018), who discovered conserved periodicity patterns across a broad range of cancers. If this was wrong, and the localisation was different in every tissue/cancer, no such patterns would have emerged in the original studies and in ours.

Generally it is an easy criticism for any analysis that it doesn't use the most up to date annotation as these are ever evolving. For example, the PCAWG consortium used the GRCh37 genome build although it was superseded by the GRCh38 genome build from 5 years earlier. The true question is whether the annotation used introduces errors and false conclusions, which to the best of our knowledge is not the case in our analysis.

If the consensus was violated in a particular cancer type, then the algorithm would need to produce a new mutational signature to explain the discrepancy. Hence such artefacts would emerge as being highly tissue specific — something that we do not observed at the selected solution. To touch upon the concerns raised by the reviewer we added the following section to the discussion of our manuscript.

Epigenetic annotations, for example, currently exist only for a subset of cancer types — and it may even be that individual cancer subtypes derive from distinct cells of origin with unique epigenomic characteristics. To facilitate a pan-cancer analysis, TensorSignatures uses, similar to previous studies (Haradhvala et al. 2016; Pich et al. 2018), consensus annotations comprising those genomic regions found to vary only lowly between different cell types. This reduces the number of annotated variants by approximately 30% as many parts of the genome are annotated as ‘variable’ (and extra state introduced by TensorSginatures). An analysis based on partially matched tissues showed that this approach is likely to underestimate the effect of genomic factors on mutagenesis and that the reduction of signal makes the analysis of rare elements, such as enhancers, noisier (Supplementary Figure 7). As more and more tissues are being genomically profiled, we expect that further tissue-specific annotations — and also entirely new genomic features – will emerge, which will produce more accurate and novel insights into the determinants of mutagenesis.

3) Nonnegative factorization of the constructed tensor

In no way, the authors’ response provides convincing evidence that TensorSignatures adds additional value as a method or “produces more accurate results” (particularly concerning statement considering the number of signatures that are not detect by TensorSignatures). As stated by the authors, performing a Tucker tensor decomposition will indeed significantly reduce the number of parameters that need to be fitted. This reduction is possible due to making assumptions of the structure of the data represented by the different tensor dimensions. Whether or not such assumptions are warranted is a philosophical question as “all models are wrong, but some are useful”. The question the authors need to answer is whether their method is useful when compared to previous approaches. Specifically, features) of mutational signatures have been reported by a number of previous studies. The authors should compare their findings to at least one such analysis. For example, somewhat similar analysis was previously reported in Nature Communications for 560 breast cancers using traditional nonnegative matrix factorization (PMID: 27136393). What is the value that TensorSignatures brings beyond such prior approaches?

A Tucker decomposition is a multidimensional generalisation of singular value decomposition. While it is a common approach for dimensionality reduction, it suffers from the same limitations as SVD or PCA in the sense that the factors have no direct biological meaning. The factorisation by TensorSignatures on the contrary extracts coefficients which have a direct interpretation: Strand-specific mutation spectra and genomic activity coefficients. For this simple reason we believe that our approach is favourable compared to generic dimensionality reduction techniques. To clarify this point in the manuscript we amended the introduction to contain the following paragraph.

This is conceptually similar to a Tucker decomposition, a multidimensional generalisation of a single value decomposition. Yet it has the advantage over the latter that the extracted factors each have a defined biological meaning.

TensorSignatures largely recovers the findings reported in the quoted study — including transcriptional and replication strand bias for multiple signatures including APOBEC. Our analysis adds to these findings of APOBEC mutagenesis by showing that two distinct APOBEC signatures with nearly identical SBS spectra, but divergent SV and replication properties exist, which are likely to be related to APOBEC3A and 3B.

TensorSignatures does so conveniently by running a single command. In addition, it sheds new light on how the signatures of UV and tobacco smoking associated mutagenesis are subtly changed across the genome - likely as a consequence of nucleotide excision repair. In fact it is these two examples that highlight the benefit of an integrated analysis as the similarity of the mutation spectra otherwise leads to a confusion of each signature's genomic properties, including strand bias. Further TensorSignatures automatically recovers many findings from other studies, as detailed in the second half of the manuscript. We believe that this sufficiently proves the utility of our tool.

4) Implementation of the tool

The implementation of the tool and its documentation are much improved. I have been able to install TensorSignatures and to run it.

Thank you for this comment.

Reviewer #3 (Remarks to the Author):

The authors have addressed my comments and have improved the manuscript in the new revision. As a minor comment around presentation, I thought that the dichotomy of discovery (PCAWG) and validation (HMF) of various signatures could be explained more clearly throughout the results section. Specific statements in the text could be added to support findings of signatures and help future readers. I also felt that the manuscript was dense and long but I will leave that up to the discussion in the other where specific comments around novelty are discussed.

Thank you for this comment.

Reviewer #4 (Remarks to the Author):

Draft comments on NCOMMS-20-30726A - Learning mutational signatures and their multidimensional genomic properties with TensorSignatures
Rozen, 6 Dec 2020

I was asked to provide an opinion on whether responses to several concerns raised by Reviewer 2 were adequate. I did not do a full review.

In summary, with one exception, I believe the responses to the Reviewer's concerns were adequate.

In a larger context, I would point out that, for discovery (extraction) of mutational signatures, the aphorism "All models are wrong, but some are useful" very much applies, as was also noted by Reviewer 2. TensorSignatures introduces a new class of models that represent a substantial conceptual advance, and the analyses reported in the manuscript indicate that these models are likely to be useful.

All methods for discovery of mutational signatures generate imperfect models and require *substantial* human interpretation in the light of all available evidence. None of the methods for discovering mutational signatures can automatically and algorithmically provide a unique "true" model. They can only provide one of many possible models that need to be further assessed.

In view of this, I am not concerned about signatures in Alexandrov et al. 2020 (<https://www.nature.com/articles/s41586-020-1943-3>) that were not detected by TensorSignatures. This is for two reasons. (1) In Alexandrov et al. and in COSMIC, some signatures were indeed pulled in from tumors that were not in the core PCAWG "platinum" data set of ~2780 genomes that the Vöhringer et al. manuscript analyzes. (2) Unsupervised learning approaches (including all methods for mutational signature discovery) require a tradeoff between sensitivity and specificity. Indeed, assessing this tradeoff is part of the human judgement needed for interpreting the results of these methods. I believe that the TensorSignatures analysis struck a reasonable balance between sensitivity and specificity.

Regarding testing against the synthetic data in Alexandrov et al., I concur with the authors that these synthetic data are not suitable: they do not have the genomic location information that TensorSignatures uses.

Regarding the Reviewer's concern that the epigenomic annotations were not as comprehensive or as up-to-date as possible, I believe that the annotations that were used are sufficient. There will always be new and better annotations. The information that can be added by these should be explored by the authors or others in the future.

Thank you for these comments.

Regarding the Reviewer's concern about comparison to `_results_` from previous approaches to understanding the relationships between signatures and genomic features, I do think a qualitative discussion of the similarities and differences to what was found in e.g. Morganella et al., Nat. Commun. 7, 11383 and/or other genomic topography/landscape papers cited in the intro would add to the manuscript. It is difficult for the reader to make comparisons between TensorSignature's findings and those the previous literature for a variety of reasons. Granted that there are some challenges, including splits / merges of SBS signatures between the current manuscript and signatures used in previous studies. Nevertheless, this discussion would add to the manuscript. I do not think TensorSignatures has to be superior in every respect to previous approaches to be useful, but I do think something in the Discussion that summarizes TensorSignature's findings in the context of findings by other results would be useful to the reader.

Thank you for this comment. To highlight how our work relates to previous works from Morganello et al. and other studies, we added the following section to the discussion:

Measuring how mutational spectra are influenced by their associated genomic features sheds light on the mechanisms underlying mutagenesis, as demonstrated by multiple previous investigations (Morganello et al. 2016; Haradhvala et al. 2016; Tomkova et al. 2018). However, as such calculations have been carried out after defining mutational signatures, they cannot detect more subtle signature changes associated with genomic features and struggle to localise very similar mutational signatures. A joint inference also helps to dissect mutational processes in situations where mutation spectra are very similar, such that genomic associations cannot be unambiguously attributed based on the mutation spectrum alone.

Minor comments: Please specify the version of SigProfiler used and any non-default parameters.

Typos: references still in the old Cell style on lines 944 and 1351.

We specified the used SigProfiler version and corrected the citation styles.

Signed: Steven G Rozen